# Explaining Vision-Language Similarities in Dual Encoders with Feature-Pair Attributions

## Abstract

Dual encoder architectures like CLIP models map two types of inputs into a shared embedding space and learn similarities between them. However, it is not understood *how* such models compare their two inputs. Similarity depends on feature-interactions rather than individual features. Here, we first derive a method to attribute predictions of any differentiable dual encoder onto feature-pair interactions between its inputs. Second, we apply our method to CLIP models and show that they learn fine-grained correspondences between parts of captions and regions in images. They match objects across input modes and also account for mismatches. This visual-linguistic grounding ability, however, heavily varies between object classes, depends on the training data distribution, and largely improves upon in-domain training. Using our method, we can identify individual failure cases and knowledge gaps about specific classes.

## 1 Introduction

Dual encoder models use independent modules to represent two types of inputs in a common embedding space and compute their similarity. The training objective is typically a triplet or contrastive loss (Sohn, 2016; van den Oord et al., 2019). Popular examples include Siamese transformers for text-text pairs (SBERT) (Reimers & Gurevych, 2019) and CLIP models (Radford et al., 2021; Jia et al., 2021) for text-image pairs. The learned representations have proven to be highly informative for downstream applications such image classification (Zhang et al., 2022a), visual question answering (Antol et al., 2015), image captioning and visual entailment (Shen et al., 2021), as well as text (Chen et al., 2023a; Yu et al., 2022) and image generation Rombach et al. (2022).

However, our understanding of which properties of the inputs these models base their predictions on is very limited. Similarities depend on interactions between two instances rather than on either instance's properties alone. Few works have studied these interactions in symmetric Siamese encoders (Eberle et al., 2020; Möller et al., 2023; 2024; Vasileiou & Eberle, 2024) and, to the best of our knowledge, they are yet to be explored in non-symmetric models like vision-language dual encoders, e.g. CLIP. First-order feature attribution methods like Shapley values (Lundberg & Lee, 2017) or integrated gradients (Sundararajan et al., 2017) are insufficient for explaining similarities, as they can only attribute predictions to individual features, not to interactions between them (Zheng et al., 2020; Ramamurthy et al., 2022).

We address this research gap by extending previous work for language-only Siamese models in NLP (Möller et al., 2023; 2024). Our contributions are: (1) We derive a method to compute general feature-pair attributions that can explain interactions between inputs of any differentiable dual encoder model. The method requires no modification of the trained model. (2) We apply the method to a range of CLIP models and show it can capture fine-grained interactions between parts of captions and corresponding regions in images as exemplified in Figure 2. It can also point out correspondence between two captions or two images (Figures 11 and 12). (3) We utilize image-captioning datasets containing object bounding-box annotations to evaluate the extent and the limit of the models' intrinsic visual-linguistic grounding abilities.

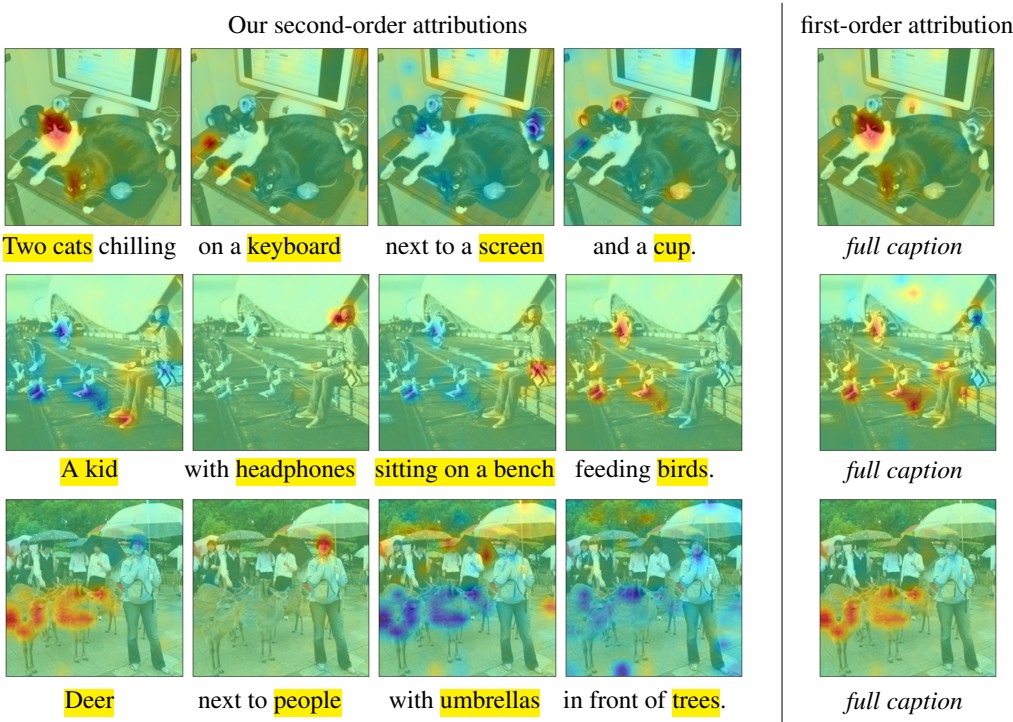

Figure 1: Examples of our second-order attributions for interactions between parts of caption and image regions vs. the analogous first-order attributions for the similarity between the image and the full caption.

## 2    RELATED WORK

**Local feature attribution methods**    aim at explaining a given prediction by assigning contributions to individual input features (Murdoch et al., 2019; Doshi-Velez & Kim, 2017; Lipton, 2018; Atanasova et al., 2020). First-order gradients can approximate a prediction's sensitivity to such features (Li et al., 2016). In transformer architectures, attention weights were proposed as explanations (Abnar & Zuidema, 2020), but ultimately rejected as only one part of the model(Jain & Wallace, 2019; Wiegreffe & Pinter, 2019; Bastings & Filippova, 2020). Layer-wise relevance propagation (LRP) defines layer-specific rules to back-propagate attributions to individual features (Montavon et al., 2019; Bach et al., 2015). In contrast shapley values (Lundberg & Lee, 2017) and IG (Sundararajan et al., 2017) treat models holistically and can provide a form of theoretical guaranty for correctness. This has recently been challenged by Bilodeau et al. (2024) who prove fundamental limitations of attribution methods. A widely used attribution method in the vision domain is GradCam (Selvaraju et al., 2017), which Chefer et al. (2021) and Bousselham et al. (2024) extend to transformer architectures. In Appendix G we discuss the relation between IG, GradCam and our work.
First-order attribution methods including the above, cannot capture dependencies on feature interactions. Tsang et al. (2018) have proposed to detect such interactions from weight matrices in feed-forward neural netorks, the Shapley value has been extended to the Shapley Interaction Index (Grabisch & Roubens, 1999; Sundararajan et al., 2020; Fumagalli et al., 2024) and Janizek et al. (2021) have generalized IG to integrated Hessians. A special case are dual and Siamese encoders whose predictions *only* depend on feature interactions between embeddings of two inputs coming from independent encoders (cf. Eq. 1). Plummer et al. (2020) and Zheng et al. (2020) have assessed image similarities. Eberle et al. (2020) have extended LRP for this model class (Vasileiou & Eberle, 2024). Möller et al. (2023) have extended IG to Siamese language encoders (Möller et al., 2024). Here we further generalize this work to multi-modal dual encoders.

**CLIP explainability.**    A number of works have focused on better understanding how CLIP models and contrastive image encoders function. Gandelsman et al. (2023) identify functions of individual

attention heads in CLIP's image encoder. Tu et al. (2024) investigate safety objectives in CLIP models and Mayilvahanan et al. (2024) analyze their out-of-domain generalization. Several works have utilized local attributions on CLIP encoders. Zhao et al. (2024) have tested a wide range of first-order methods attributing similarity scores onto images and captions. With the InteractionCAM baseline, Sammani et al. (2023) have proposed a method to assess feature interactions in contrastive models. InteractionLIME has pioneered the attribution of interactions between captions and images in CLIP models (Joukovsky et al., 2023), and as such is the closest related method to ours. However, it bi-linearly approximates CLIP and, therefore, cannot explain the actual model. Next to gradient-based attribution methods, Li et al. (2022c; 2023) and Black et al. (2022) have proposed forward-facing saliency methods for similarity models.

## 3 METHOD

We first derive general feature-pair attributions for dual encoder predictions and then specifically apply the result to vision-language models.

**Derivation of interaction attributions.**    Let

$$s = f(\mathbf{a}, \mathbf{b}) = \mathbf{g}(\mathbf{a})^T \mathbf{h}(\mathbf{b}) \tag{1}$$

be a differentiable dual-encoder model, with two vector-valued encoders $\mathbf{g}$ and $\mathbf{h}$, respective inputs $\mathbf{a}$ and $\mathbf{b}$ and a scalar output $s$. For our purpose, $\mathbf{g}$ will be an image encoder with an image input $\mathbf{a}$ and $\mathbf{h}$ will be a text encoder with a text representation $\mathbf{b}$ as input. To attribute the prediction $s$ onto features of the two inputs $\mathbf{a}$ and $\mathbf{b}$, we also define two uninformative *reference* inputs $\mathbf{r}_a$, the black image, and $\mathbf{r}_b$, a sequence of padding tokens with fixed length. We then rigorously start from the following expression:

$$f(\mathbf{a}, \mathbf{b}) - f(\mathbf{r}_a, \mathbf{b}) - f(\mathbf{a}, \mathbf{r}_b) + f(\mathbf{r}_a, \mathbf{r}_b) \tag{2}$$

Our derivation first proceeds by showing the equality of this initial starting-point to Eq. 10. We then reduce this equality to our final attributions in Eq. 11 using the approximations discussed below. As a first step, seeing $f$ as an anti-derivative, we can turn the above formula into an integral over its derivative:

$$\left[ f(\mathbf{a}, \mathbf{b}) - f(\mathbf{r}_a, \mathbf{b}) \right] - \left[ f(\mathbf{a}, \mathbf{r}_b) - f(\mathbf{r}_a, \mathbf{r}_b) \right]$$

$$= \int_{\mathbf{r}_b}^{\mathbf{b}} \frac{\partial}{\partial \mathbf{y}_j} \left[ f(\mathbf{a}, \mathbf{y}) - f(\mathbf{r}_a, \mathbf{y}) \right] d\mathbf{y}_j = \int_{\mathbf{r}_b}^{\mathbf{b}} \int_{\mathbf{r}_a}^{\mathbf{a}} \frac{\partial^2}{\partial \mathbf{x}_i \partial \mathbf{y}_j} f(\mathbf{x}, \mathbf{y}) \, d\mathbf{x}_i \, d\mathbf{y}_j \tag{3}$$

Here, $\mathbf{x}$ and $\mathbf{y}$ are integration variables for the two inputs. We use component-wise notation with the indices $i$ and $j$ for the input dimensions and omit sums over double indices for clarity. We plug in the model definition from Equation 1:

$$\int_{\mathbf{r}_a}^{\mathbf{a}} \int_{\mathbf{r}_b}^{\mathbf{b}} \frac{\partial^2}{\partial \mathbf{x}_i \partial \mathbf{y}_j} \mathbf{g}_k(\mathbf{x}) \, \mathbf{h}_k(\mathbf{y}) \, d\mathbf{x}_i \, d\mathbf{y}_j \tag{4}$$

Again, we use component-wise notation for the dot-product between the two embeddings $\mathbf{g}(\mathbf{x})$ and $\mathbf{h}(\mathbf{y})$ and index output dimensions with $k$. Since neither embedding depends on the other integration variable, we can separate the integrals:

$$\int_{\mathbf{r}_a}^{\mathbf{a}} \frac{\partial \mathbf{g}_k(\mathbf{x})}{\partial \mathbf{x}_i} \, d\mathbf{x}_i \int_{\mathbf{r}_b}^{\mathbf{b}} \frac{\partial \mathbf{h}_k(\mathbf{y})}{\partial \mathbf{y}_j} \, d\mathbf{y}_j \tag{5}$$

This step makes explicit use of the strict independence of the two encoders. Cross-encoder architectures would introduce dependencies between them. Both terms are line integrals from the references to the actual inputs in the respective input representation spaces; $\partial \mathbf{g}_k(\mathbf{x})/\partial \mathbf{x}_i$ and $\partial \mathbf{h}_k(\mathbf{y})/\partial \mathbf{y}_j$ are the Jacobians of the two encoders. Following the concept of integrated gradients (Sundararajan et al., 2017), we define the straight lines between both references and inputs,

$$\mathbf{x}(\alpha) = \mathbf{r}_a + \alpha(\mathbf{a} - \mathbf{r}_a), \tag{6}$$

$$\mathbf{y}(\beta) = \mathbf{r}_b + \beta(\mathbf{b} - \mathbf{r}_b), \tag{7}$$

parameterized by $\alpha$ and $\beta$, and solve by substitution. For the integral over encoder $\mathbf{g}$ this yields

$$\int_0^1 \frac{\partial \mathbf{g}_k\left(\mathbf{x}(\alpha)\right)}{\partial \mathbf{x}_i} \frac{\partial \mathbf{x}_i(\alpha)}{\partial \alpha}\, d\alpha \;=\; (\mathbf{a} - \mathbf{r}_a)_i \int_0^1 \frac{\partial \mathbf{g}_k\left(\mathbf{x}(\alpha)\right)}{\partial \mathbf{x}_i}\, d\alpha, \tag{8}$$

since $\partial \mathbf{x}(\alpha)/\partial \alpha = (\mathbf{a} - \mathbf{r}_a)$, which is a constant w.r.t $\alpha$; hence, we can pull it out of the integral. The integral over encoder $\mathbf{h}$ is processed in the same way. We then define the two *integrated Jacobians*,

$$\mathbf{J}_{ki}^a = \int_0^1 \frac{\partial \mathbf{g}_k\left(\mathbf{x}(\alpha)\right)}{\partial \mathbf{x}_i}\, d\alpha \approx \frac{1}{N} \sum_{n=1}^{N} \frac{\partial \mathbf{g}_k(\mathbf{x}(\alpha_n))}{\partial \mathbf{x}_i}, \tag{9}$$

and $\mathbf{J}_{kj}^b$ analogously. In practice, these integrals are calculated numerically by sums over $N$ steps, with $\alpha_n = n/N$. This introduces an approximation error which must, however, converge to zero for large $N$ by definition of the Riemann integral. We plug the results from Equation 8 and the definitions of the *integrated Jacobians* back into Equation 5 and obtain:

$$(\mathbf{a} - \mathbf{r}_a)_i\, \mathbf{J}_{ik}^a \mathbf{J}_{kj}^b\, (\mathbf{b} - \mathbf{r}_b)_j =: \mathbf{A}_{ij} \tag{10}$$

After computing the sum over the output embedding dimensions $k$, this provides a matrix of interactions between feature-pairs $(i, j)$ in input $\mathbf{a}$ and $\mathbf{b}$ which we call attribution matrix $\mathbf{A}_{ij}$. Note that except for the numerical integration, the equality to Equation 2 still holds. Hence, the sum over all feature-pair attributions in $\mathbf{A}$ is an exact reformulation of the ansatz. If the references $\mathbf{r}_a$ and $\mathbf{r}_b$ are uninformative, i.e. $f(\mathbf{r}_a, \mathbf{b}) \approx 0$, $f(\mathbf{a}, \mathbf{r}_b) \approx 0$, $f(\mathbf{r}_a, \mathbf{r}_b) \approx 0$, we arrive at the final approximation

$$f(\mathbf{a}, \mathbf{b}) \approx \sum_{ij} \mathbf{A}_{ij}, \tag{11}$$

where $i$ ranges over dimensions in input $\mathbf{a}$ and $j$ over $\mathbf{b}$. This provides an approximate decomposition of the model prediction $s = f(\mathbf{a}, \mathbf{b})$ into additive contributions of feature-pairs in the two inputs.

**Inter-modal attributions.** In the derivation above, we treat image and text representations as vectors. In current transformer-based language encoders, text inputs are represented as $S \times D_b$ dimensional tensors, where $S$ is the length of the token sequence and $D$ is the model's embedding dimensionality. In vision transformers, image representations are $P \times P \times D_a$ dimensional tensors, with $P$ being the number of patches that the image is split into horizontally and vertically. Our pair-wise image-text attributions thus have the dimensions $P \times P \times D_a \times S \times D_b$. With hundreds of embedding dimensions and tens of tokens and patches, this quickly becomes intractable. Fortunately, the sum over dimensions in Equation 11 enables the additive combination of attributions in $\mathbf{A}$. We sum over the embedding dimensions of both encoders $D_a$ and $D_b$ and obtain a $P \times P \times S$ dimensional attribution tensor, which estimates for *each pair of a text token and an image patch* how much their combination contributes to the overall prediction. These attributions are still three-dimensional and thus not straightforward to visualize. However, again we can use their additivity, slice the 3d attribution tensor along text or image dimensions and project onto the remaining dimensions by summation. We can for example select a slice over a range of tokens and project it onto the image as in Figure 2 (a)/(b). Attribution heat maps over the image result from interpolating the patch-level attributions to image resolution. The reverse case of slicing parts of the image and projecting the result onto the caption dimension is shown in Figure 2(c)/(d). Here, we project the selected image slices marked by yellow bounding boxes onto the caption and visualize attributions as saliency maps over tokens in the caption.

**Intra-modal attributions.** Albeit vision-language dual-encoders are typically trained to match images against captions, we can compute attributions for image-image or text-text pairs as well by applying the same encoder to both inputs. In Appendix A, we describe this in more detail and show examples.

## 4 EXPERIMENTS

In the experiments, we apply our feature-pair attributions to contrastively trained vision-language dual encoders. We focus on evaluating the interactions between mentioned objects in captions and corresponding regions in images by selecting sub-strings in captions and analyzing their interactions with image patches, as illustrated in Figure 2 (a)/(b). Throughout our experiments, we attribute to the second-last hidden representation in the models' image and text encoder and use $N = 50$.

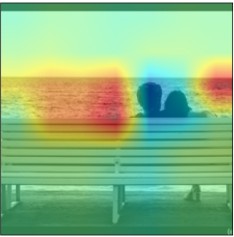 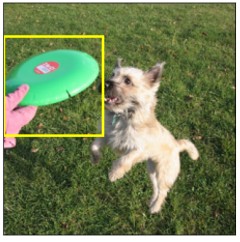 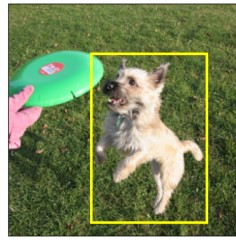

(a) A couple sitting on a bench looking at the sea.

(b) A couple sitting on a bench looking at the sea.

(c) A dog is jumping for a frisbee.

(d) A dog is jumping for a frisbee.

Figure 2: Inter-modal attributions between: (left) selected parts of a caption in yellow and an image, heatmaps over the image are red for positve and blue for negative; (right) selected bounding-boxes in the image and the caption, saliencies over captions are red for positive and blue for negative.

**Datasets.** We base this evaluation on three image-caption datasets that also contain object bounding-box annotations in images, Microsoft's COCO (Lin et al., 2014), the Flickr30k collection (Young et al., 2014; Plummer et al., 2015), and the HNC dataset by Dönmez et al. (2023). For HNC, we follow their approach by generating captions from scene graphs using templates. Specifically, we use a basic template of the form "<subject> <predicate> <object>" to align the generated captions with the domain of the other two datasets. In our analysis, we use HNC for evaluation only, on Flickr30k we use the test split, and on COCO we use the validation split as the test split does not contain captions[1].

**Models.** We analyze CLIP dual-encoder architectures (Radford et al., 2021) without cross-encoder dependencies and the standard inter-modal contrastive objective. We evaluate the original OpenAI models, as well as the OpenCLIP reimplementations trained on the *Laion* (Schuhmann et al., 2022), *Dfn* (Fang et al., 2024), *CommonPool* and *DataComp* (Gadre et al., 2023) datasets, as well as *MetaCLIP* (Xu et al., 2024) [2].

**Fine-tuning.** Next to the unmodified models, we evaluate variants fine-tuned on the COCO and Flickr30k train splits. All trainings run for five epochs using the AdamW optimizer (Loshchilov & Hutter, 2018) with an initial learning rate of $1 \times 10^{-7}$, exponentially increasing to $1 \times 10^{-5}$, a weight decay of $1 \times 10^{-4}$, and a batch size of $64$ on one NVIDIA A6000 GPU.

### 4.1 OBJECT LOCALIZATION

To systematically assess the visual-linguistic grounding abilities of the analyzed dual encoders, we evaluate the models' localization ability of objects in the image that are mentioned in a caption. For this experiment, we use all object annotations for which a single instance of its class appears in the image and its bounding-box is larger than one patch. For COCO, we identify class occurrences in the caption through a dictionary based synonym matching. For HNC, classes exactly match sub-strings in captions and in Flickr30k, respective spans are already annotated. This results in 3.5k image-caption pairs from COCO, 8k pairs from Flickr30k, and 500 pairs from HNC.

**Localization evalutaion.** We compute attributions between the token range of a class mention in a given caption and the image. Following Zhao et al. (2024), we then employ the Point Game (PG) framework by Zhang et al. (2018) to evaluate how well attributions correspond to human bounding-box annotations. It defines PG Accuracy (PGA) as the fraction of cases for which the maximally attributed patch lies within the objects' bounding-box, and PG Energy (PGE) as the fraction of positive attributions falling inside a given bounding-box over the total attribution to the full image (Zhao & Chan, 2023; Wang et al., 2020). Figure 3 shows examples from different PGE-ranges. Very high or low values, unambiguously indicate object correspondence or clear failure cases, respectively (examples a and d). Intermediate values, however, often arise from attributions

---

[1]https://www.kaggle.com/datasets/shtvkumar/karpathy-splits

[2]CLIP family: https://github.com/openai/CLIP, Open family: https://github.com/mlfoundations/open_clip

extending to the context beyond the bounding box, such as the *shirt* in (b) and the *tennis court* in (c). Figure 4 (Left) shows cumulative $PGE$-distributions for the OpenClip models on the COCO dataset. Table 1 shows the median PGE and PGA for the OpenAI model and the OpenClip Laion counterpart both implementing the ViT-B-16 architecture on all three evaluation datasets. Results for all tested OpenAI and OpenCLIP models on all datasets are included in Appendix C.

**In-domain fine-tuning.**  The tested models are trained with large captioning datasets from the web but have (presumably) not been tuned on the Flickr30k and COCO train splits. In order to assess domain-effects of the models' grounding ability, we fine-tune them on the respective train splits. We emphasize that all fine-tunings are performed in the standard contrastive setting and never change model architectures nor training objectives to explicitly perform grounding. We then compare the grounding ability of unmodified models and their fine-tuned counterparts by assessing their full $PGE$-distributions and test whether the stochastic dominance of one over the other is significant (Dror et al., 2019) (details in Appendix F).
For both the OpenAI and OpenCLIP models, fine-tuning increases grounding abilities by a large margin. These improvements are consistently significant at a strict criterion of $p < 0.001$ and $\epsilon = 0.01$. While the unmodified CLIP ViT-B/16 model already has good grounding abilities on COCO and Flickr30k (the examples in Figure 3 are from this model), the off-the-shelf OpenCLIP counterparts ground rather poorly on these datasets. However, their improvement upon in-domain fine-tuning is remarkable, which is apparent in Figure 7. The off-the-shelf model cannot identify the *clock* and even attributes the *surfboard* negatively, while the fine-tuned version points out both clearly.

**Class-wise evaluation.**  To test the knowledge about specific visual-linguistic concepts in the models and how it changes upon in-domain tuning, we can break down the above analysis for individual classes. Figure 5 shows average $PGE$-values and their standard deviations in the OpenClip Dfn model for all COCO classes, ordered from left to right by how good their average grounding ability is in the unmodified model (blue). Values for $PGE$ range from $0.92 \pm 0.08$ for *sheep* to $0.07 \pm 0.07$ for *snowboard*. The model can already point out the leftmost classes *sheep*, *bear*, *elephant* etc. very well, while for the rightmost classes *snowboard*, *cell phone*, *baseball bat*, etc., intrinsic grounding is poor. Upon fine-tuning (orange) most classes improve. Using the standardized mean difference of the two $PGE$-values as a measure for effect size, we observe the largest improvements for the classes *horse*, *bench*, *giraffe*, *airplane* and *clock*. In Appendix C (Figure 19), we repeat this experiment for the Laion and CommonPool models and observe similar results.

**Baselines.**  To test whether the complexity of our attributions is necessary to assess the models' interactions between captions and images, we compare our method against two baselines: the InteractionCAM by Sammani et al. (2023) extending GradCAM to contrastive encoders (cf. Appendix G), and the ITSM method by Li et al. (2022c) resulting from pair-wise multiplication of token and patch embeddings. In contrast to other first-order attribution methods, both of these baselines can assess interactions between captions and images and, like our method, do not modify model parameters, embeddings nor gradients. Figure 4 (right) includes cumulative $PGE$-distributions for our attributions and both baselines, for a selection of models on the COCO test split. Our method results in significantly better $PGE$-statistics ($p < 0.001$, $\epsilon = 0.01$). Table 2 shows median PGE and PGA on COCO and Flickr30k for the OpenAI model, quantifying the large performance margin between our method and the two baselines. Results for the Laion, DataComp and DFN model are included in Table 6. Figure 15 shows qualitative comparison of the interaction attributions by the three methods.

### 4.2 OBJECT DISCRIMINATION

In many of the above examples, we observe that attributions between a given object in the text and a non-matching one in the image or vice versa are often not only neutral but negative. Figure 9 includes four explicit examples. For a systematic evaluation of this effect, we sample instances from COCO that include at least two different object classes, which both appear exactly once in the image and are mentioned in the caption. We compute attributions between the two corresponding bounding-boxes and text spans and also across them, which we refer to as cross-attribution. The attribution to the actual object's bounding-box is almost always positive (97.1%), while cross-attributions to the other object are negative in 65.6% of the cases (70.1% in the COCO fine-tuned model - cf. Figure 16 in

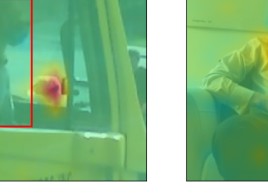 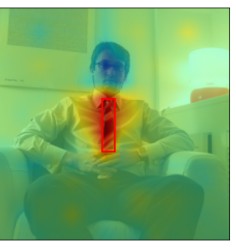 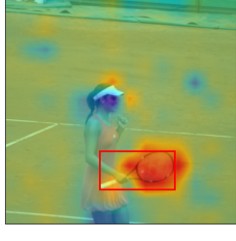 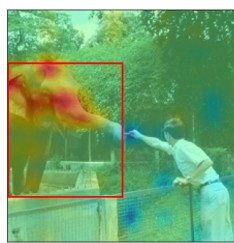

(a) Dog sitting inside the cab of a yellow truck. ($PGE\!=\!0.1$)

(b) A man in a shirt and tie sitting on a white chair. ($PGE\!=\!0.4$)

(c) A woman on the court while holding her tennis racket. ($PGE\!=\!0.5$)

(d) A man is feeding an elephant over a fence. ($PGE\!=\!0.9$)

Figure 3: Examples for attribuitions between selected objects in the caption (yellow) and the image together with corresponding COCO bounding-boxes. $PGE$ is the fraction of positive attributions falling inside the box as described in Section 4.1.

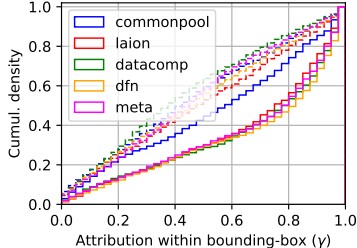 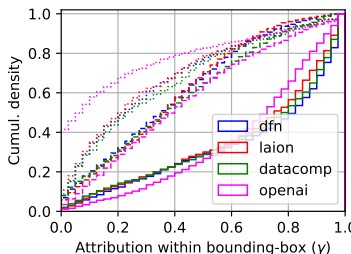

Figure 4: (Left) Cumulative distributions of the fraction of attributions falling within corresponding COCO object bounding-boxes ($PGE$) as described in Section 4.1 before (dashed) and after (solid) in-domain fine-tuning. (Right) Cumulative $PGE$-distributions for our method (solid), the InteractionCAM (dashed) and ITSM (dotted) baselines.

Table 1: Summary of the vision-language grounding evaluation for the ViT-B-16 models trained by OpenAI and on Laion. *Tuning* is whether the model was fine-tuned on the train split of the respective dataset, mPGE reports median Point Game Energy and PGA is the Point Game Accuracy. Full results in Table 5 and 4.

| Training | Tuning | COCO | | HNC | | Flickr30k | |
|---|---|---|---|---|---|---|---|
| | | mPGE | PGA | mPGE | PGA | mPGE | PGA |
| OpenAI | No | 72.3 | 79.0 | 57.0 | 65.0 | 64.4 | 72.1 |
| | Yes | 78.0 | 82.9 | - | - | 73.4 | 79.0 |
| Laion | No | 49.4 | 63.3 | 40.0 | 51.6 | 38.2 | 52.0 |
| | Yes | 71.1 | 83.2 | - | - | 54.6 | 61.8 |

Table 2: Point Game comparison of our attributions against the ITSM method and InteractionCAM (ICAM) for the OpenAI and Laion model. Results for more models are in Table 6.

| Training | Method | COCO | | Flickr30k | |
|---|---|---|---|---|---|
| | | mPGE | PGA | mPGE | PGA |
| OpenAI | ITSM | 18.1 | 21.4 | 19.5 | 23.3 |
| | ICAM | 38.6 | 54.6 | 33.5 | 51.4 |
| | Ours | **72.3** | **79.0** | **64.4** | **72.1** |
| Laion (tuned) | ITSM | 22.8 | 30.3 | 24.5 | 28.7 |
| | ICAM | 32.5 | 58.4 | 33.5 | 51.4 |
| | Ours | **71.1** | **83.2** | **54.3** | **61.8** |

Appendix C). This shows that the models do indeed often attribute mis-matching objects negatively, however, this is not consistently the case.

We further investigate cases where cross-attribution tends to be positive. Table 7 (Appendix C) shows the five class-pairs with the highest average values. All are between classes that often occur together in the labels often involving people but also *Toilet* and *Sink*. We hypothesize that the models may positively relate objects that frequently appear together and test it by correlating class co-occurrence in the labels with average cross-attribution between all classes. The Spearman (Pearson) correlation is $r_S = 0.25$ ($r_P = 0.33$), indicating that class co-occurrence may moderately affect positive cross-attribution but is likely not the only reason for it.

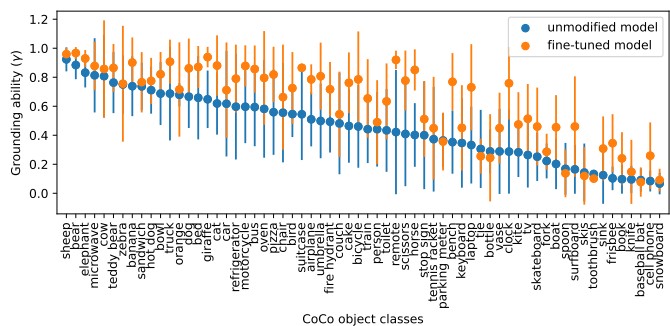

Figure 5: Class-wise average Point Game Energy ($PGE$) and its standard deviation (error bars) of the OpenClip DFN model before and after in-domain fine-tuning on the COCO train split.

Figure 6: Decline of similarity scores between images and captions for iterative conditional image patch deletions.

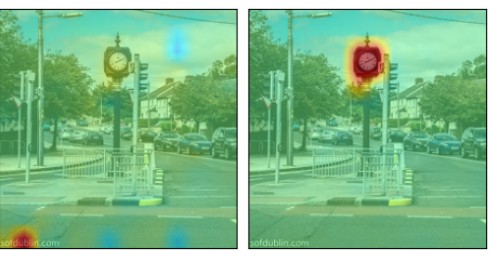

(a) A clock on a pole in the intersection of two streets.

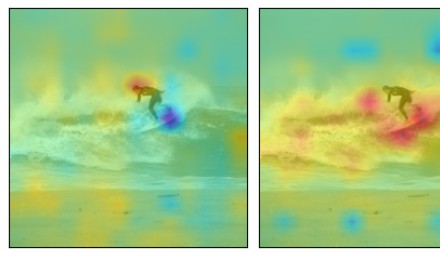

(b) A surfer riding a wave on a yellow surfboard.

Figure 7: Attribution differences between the off-the-shelf OpenCLIP Laion model (left image) and a version fine-tuned on COCO (right image). Attribution heat maps are for selected caption parts in yellow.

## 4.3 INPUT PERTURBATION

**Insertion and Deletion.** We evaluate the attribution quality through a perturbation experiment (Samek et al., 2016). We follow Sammani et al. (2023) to extend perturbation evaluations to contrastive models by conditionally removing or inserting the most attributed features in one input while keeping the other input unmodified and compare our method with random selection, InteractionCAM, and ITSM baselines. Figure 6 plots the decrease in similarity score for conditional image patch deletion (CID). Our method produces the steepest score decline as a function of the number of patches removed, indicating its ability to identify the most relevant interactions. Next to CID, we also evaluate image patch insertion (CII) as well as conditional text token deletion/insertion (CTD/CTI). All plots

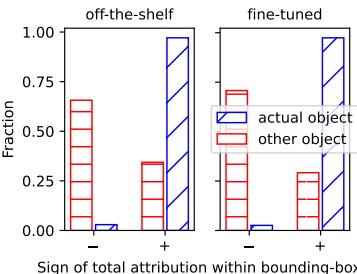

Figure 8: Distribution of signs of attributions to actual and other objects in the image as described in Section 4.2.

| Method | CID ↓ | CII ↑ | CTD ↓ | CTI ↑ |
|--------|-------|-------|-------|-------|
| Ours | **64** | **112** | **6.4** | **7.4** |
| Random | 83 | 84 | 6.6 | 6.8 |
| ICAM | 89 | 80 | 6.5 | 6.9 |
| ITSM | 99 | 69 | 6.8 | 6.7 |

Table 3: AUC for conditional image deletion (CID), conditional image insertion (CII), conditional text deletion (CTD), and conditional text insertion (CTI), performed on COCO for models pre-trained on Laion and fine-tuned on COCO. ↓ denotes lower is better and ↑ denotes greater is better. Corresponding plots are in Figure 20.

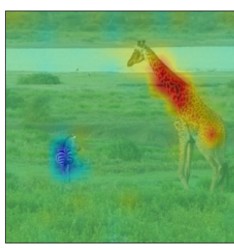
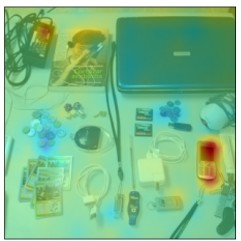
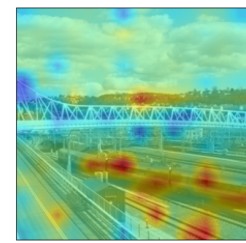
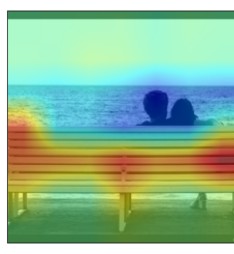

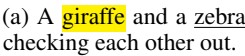

(a) A ==giraffe== and a zebra checking each other out.

(b) A desk with items including a ==cellphone== and pocket change.

(c) A long bridge with a ==train== going underneath.

(d) A couple ==sitting on a bench== looking at the sea.

Figure 9: Attributions between selected parts of a caption (yellow) and a corresponding image. Other objects that also appear in the image and are mentioned in the caption (underlined) but are not selected for attribution often receive negative attributions (blue parts of the attribution heatmaps).

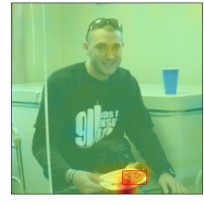
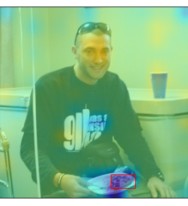
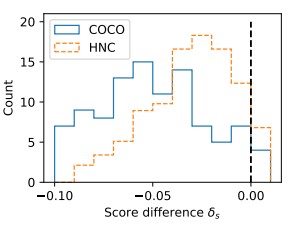
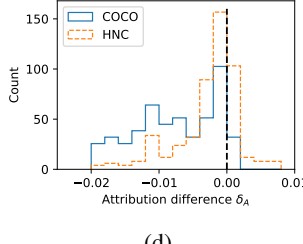

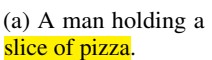

(a) A man holding a ==slice of pizza==.

(b) A man holding a ==coffee mug==.

(c)

(d)

Figure 10: (Left) Example attributions for hard negative captions. True object marked in yellow, negative in magenta. COCO bounding-boxes in red. (Right) Histograms for score ($\delta_S$) and attribution ($\delta_A$) changes in hard negative captions (cf. Section 4.3).

are included in Figure 20. Table 3 provides a summary and reports the area under the curve (AUC) for the four variants. Our method consistently results in the highest AUC values for the insertion experiments and the lowest for deletion.

**Hard negative captions.** Insertion and deletion experiments have been criticized for producing out-of-domain inputs Hooker et al. (2019). On the text side, it is straightforward to produce in-domain perturbations. We create hard negative captions that replace a single object in a positive caption with a reasonable but different object to receive a negative counterpart. To this end, we leverage the automatic procedure by (Dönmez et al., 2023) together with our simplified template (cf. Section 4) and additionally create a second resource from COCO by manually annotating a small yet high-quality evaluation sample of 100 image-caption pairs.

We check whether our negative captions actually result in a decrease of the predicted similarity score compared with their positive counterparts and define the difference as $\delta_S$. It is negative in 95.2% (89.1%) of the COCO (HNC) pairs. We then compute attributions between the token range of the original or replaced object and the object bounding-box in the image and define the attribution difference between the negative and the positive caption as $\delta_A$. It is also negative in 95.2% (74.1%) of the COCO (HNC) examples. Full histograms for $\delta_A$ and $\delta_S$ are included in Figure 10 (Right). These results show that the model mostly reacts correctly to the mistake in the caption and decreases the assigned attribution. An examples are included in Figure 10 (Left).

## 5 DISCUSSION

**Interpretation of results.** Prior to us, others have shown which areas in images and tokens in captions have an influence on CLIP similarities by means of first-order attributions. However, by enabling second-order attributions, to our knowledge, our evaluation is the first that analyzes *interactions* between captions and images. This way we can assess fine-grained correspondence

between parts of captions and regions in images.

An interesting finding is that the models do not only match objects, but can also penalize mismatches by assigning negative contributions to them. However, such cross-attributions also happen to be positive in other cases and we cannot ultimately clarify what determines their sign. Objects that frequently occur in the same context, like various things associated with people, toilets and sinks, cars and buses, etc. appear to have an influence but do not seem to be the only determining factor. We note that positive cross-attributions need not indicate the model cannot differentiate between two objects, but may be due to the presence of one object increasing the likelihood of observing another. Future work should develop a better understanding of this phenomenon.

The fact that the models' grounding ability can be poor for individual classes and improves by large margins upon in-domain tuning shows that the models can have knowledge gaps about particular classes and require explicit exposure to their visual-linguistic concepts to develop a robust inter-modal correspondence. Despite clip embeddings being known to be among the most generalizable available, this finding indicates that there is still room for improvement.

While the unmodified OpenCLIP models frequently exhibit obvious misattributions, we can hardly identify unreasonable attributions in the tuned versions nor the OpenAI models. Qualitatively in these models attributions outside of object bounding-boxes occur in visually correlated scenes like bathrooms or streets (Figure 14 (a), (c)), true misidentification in difficult contexts (e.g. the *dog* in Figure 3), partial coverage (Figure 14 (d)) or attributions falling just outside of boxes (Figure 14 (b)).

**Limitations.** Our feature-pair attributions are an approximation as Equation 11 clearly states. Moreover, throughout this work, we attribute to deep representations of inputs because it is computationally feasible and informative (Möller et al., 2024). In transformers, deep representations have undergone multiple contextualization steps and are technically not bound to input features at the given position. Last, recently proven fundamental limitations of attribution methods urge caution in their interpretation, especially regarding counterfactual conclusions about the importance of individual features for the overall prediction (Bilodeau et al., 2024). Despite these considerations, empirically, our evaluations show that our derived feature-pair attributions produce reasonable results in a large majority of cases and can point out general inabilities (misattribution before fine-tuning), individual errors (misidentification of objects), and biases (positive cross-attribution of objects). While they should not be seen as guaranteed robust and faithful explanations, we argue that our attributions do provide valuable insights into dual-encoder models and have the potential to improve these models further.

# 6 CONCLUSION

In this paper, we have derived general feature-pair attributions for dual-encoder architectures enabling the attribution of similarity predictions for two inputs onto interactions of their features. Our method applies to any differentiable dual-encoder architecture and requires no modification of the model itself, its representations or gradients. We believe it can lead to valuable insights in other applications like image similarity or (multi-modal) information-retrieval and help improve these models further. Applying our method to CLIP models shows that they learn fine-grained correspondence between visual and linguistic concepts. Mis-matching or wrong objects are often not only ignored, but contribute negatively to image-caption similarities. However, this inter-modal correspondence can be poor when models are not exposed to matching data distributions during training and we can identify knowledge gaps about specific object classes. In-domain fine-tuning can substantially improve these gaps pointing out weak spots in the generalization of the initial models.

# 7 REPRODUCIBILITY STATEMENT

Upon publication we will make our code available on GitHub. For reviewers to verify the implementation of our method, we include code of the core functionality in the supplementary material.

For the implement of our method, we make use of the auto-differentiation framework in the PyTorch package. For a give input $\mathbf{x}(\alpha_n)$, $\mathbf{g}(\mathbf{x}(\alpha_n))$ is the forward pass through the encoder $\mathbf{g}$, and the Jacobian $\partial \mathbf{g}_k(\mathbf{x}(\alpha_n))/\partial \mathbf{x}_i$ is the corresponding backward pass. For an efficient computation of all $N$ interpolation steps in Eq. 9, we can batch forward and backward passes since individual interpolations are independent of another.

In practice, we attribute to intermediate representations, thus, the interpolations in Eq. 6 are between latent representations of the references and inputs. We use PyTorch *hooks* to compute these interpolations during the forward pass.

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

A hot dog sitting on a table covered in confetti.
Surrounded by glitter, there is a sausage in a bun.

---

A hot dog sitting on a table covered in confetti.
Surrounded by glitter, there is a sausage in a bun.

Figure 11: Intra-modal text-text attributions between top and bottom captions (top: selections in yellow, bottom: saliencies as above).

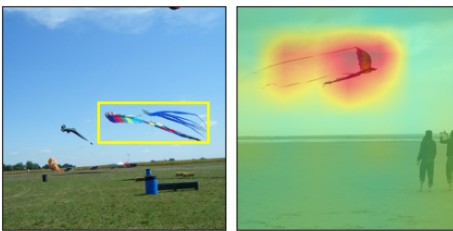

Figure 12: Intra-modal image-image attributions between left and right image (left: selection in yellow, right: heatmaps as above). More examples in Figure 13

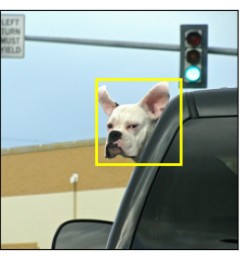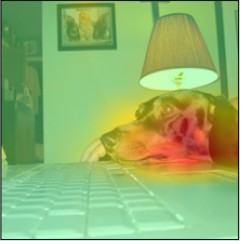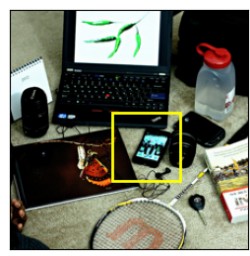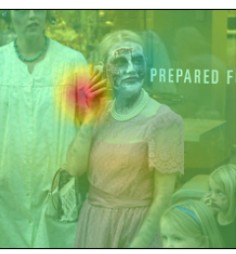

Figure 13: Image-image attributions between the yellow bounding-box in the left image and the one to its right as described in Section 3

## A   INTRA-MODAL ATTRIBUTIONS

Albeit vision-language dual-encoders are typically trained to match images against captions, we can compute attributions for image-image or text-text pairs as well by applying the same encoder to both inputs. For text-text attributions, after summation over embedding dimensions, this yields an $S_1 \times S_2$ dimensional attribution tensor, with $S_1$ and $S_2$ being token sequence lengths of the two texts. These 2d attributions may be visualized in the form of a matrix (Möller et al., 2023). In Figure 11 we, however, stick to the slice representation and attribute the yellow selected slice in the first caption onto the second caption. For image-image similarities, attribution tensors become four dimensional taking the shape $(P \times P)_1 \times (P \times P)_2$ and containing a contribution for every pair of two patches from either image. Parentheses indicate which input the dimensions belong to. In Figure 12, we attribute the slice of the yellow bounding-box in the left image onto the image to its right. Appendix B includes additional examples.

## B   ADDITIONAL EXAMPLES

Figure 13 shows two more examples for image-image attributions as described in Section 3. Figure 14 shows error cases of attributions that are discussed in Section 5.

## C   ADDITIONAL RESULTS

Figure 16 shows the distribution of attribution signs that the analysis in Section 4.2 is based on. Figure 17 and 18 provide additional plots of cumulative PGE-distributions for different models and datasets and Figure 19 shows class-wise evaluations of PGE for two additional OpenCLIP models.

## D   CROSS OBJECT ATTRIBUTION

Table 7 shows class pairs from COCO with a positive cross-attribution as discussed in Section 4.2.

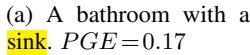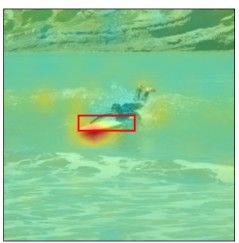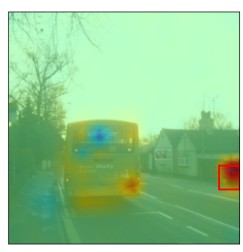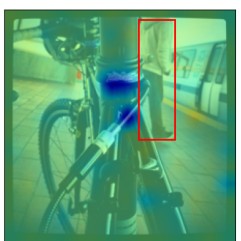

(a) A bathroom with a sink. $PGE = 0.17$

(b) Somone is going down a wave lying on a surfboard belly down. $PGE = 0.1$

(c) A yellow bus and a red car on a street. $PGE = 0.08$

(d) A waiting person behind a locked bicycle. $PGE = 0.1$

Figure 14: Different error cases with low $PGE$-values as discussed in Section 5. COCO bounding-boxes (red) are for corresponding token-ranges (yellow).

Table 4: Summary of the vision-language grounding evaluation for all OpenClip models on COCO and Flickr30k. The *Training* column refers to the dataset the model was initially trained on, *Tuning* is whether the model was addionally fine-tuned on the respective evaluation dataset. All models implement the ViT-B-16 architecture except Meta-Clip that uses quickgelu activations.

| | | COCO | | | Flickr30k | | |
|---|---|---|---|---|---|---|---|
| **Training** | **Tuning** | mPGE | PGE>0.8 | PGA | mPGE | PGE>0.8 | PGA |
| Laion | No | 49.4 | 22.0 | 63.3 | 38.2 | 15.9 | 52.0 |
| | Yes | 71.1 | 47.3 | **83.2** | **54.6** | 30.6 | **61.8** |
| CommonPool | No | 43.0 | 18.2 | 58.8 | 36.7 | 15.5 | 53.0 |
| | Yes | 57.7 | 28.7 | 67.1 | 44.6 | 20.8 | 56.2 |
| DataComp | No | 38.5 | 14.6 | 56.0 | 32.8 | 11.8 | 48.9 |
| | Yes | **72.4** | 50.0 | 75.1 | 50.7 | 27.3 | 56.0 |
| DFN | No | 46.5 | 19.6 | 54.3 | 35.4 | 12.3 | 43.3 |
| | Yes | 71.4 | **53.3** | 74.6 | 53.1 | **33.5** | 58.3 |
| Meta-Clip | No | 44.2 | 16.8 | 52.3 | 37.0 | 14.5 | 46.4 |
| | Yes | 57.5 | 49.8 | 77.1 | 49.2 | 24.1 | 57.2 |

Table 5: Summary of the vision-language grounding evaluation for different CLIP models by OpenAI as described in Section 4.1. *Model* refers to the investigated architecture, *Tuning* is whether the model was fine-tuned on the train split of the respective dataset.

| | | COCO | | | HNC | | | Flickr30k | | |
|---|---|---|---|---|---|---|---|---|---|---|
| **Model** | **Tuning** | mPGE | PGE>0.8 | PGA | mPGE | PGE>0.8 | PGA | mPGE | >0.8 | PGA |
| RN50 | No | 66.3 | 28.8 | 76.9 | 50.1 | 22.6 | 61.8 | 60.1 | 25.5 | 71.2 |
| ViT-B/32 | No | 63.5 | 33.3 | 69.1 | 52.8 | 28.5 | 58.5 | 50.4 | 23.4 | 58.1 |
| ViT-B/16 | No | 72.3 | 35.7 | 79.0 | 57.0 | 31.7 | 65.0 | 64.4 | 28.4 | 72.1 |
| ViT-B-16 | Yes | **78.0** | **48.4** | **82.9** | - | - | - | 73.4 | 40.7 | 79.0 |

# E    PERTURBATIONS

## E.1    CONDITIONAL INSERTION AND DELETION

## E.2    HARD NEGATIVE CAPTIONS

Figure 21 shows full histograms of $\delta_S$ and $\delta_A$ as discussed in the experiments with hard negative captions in Section 4.3.

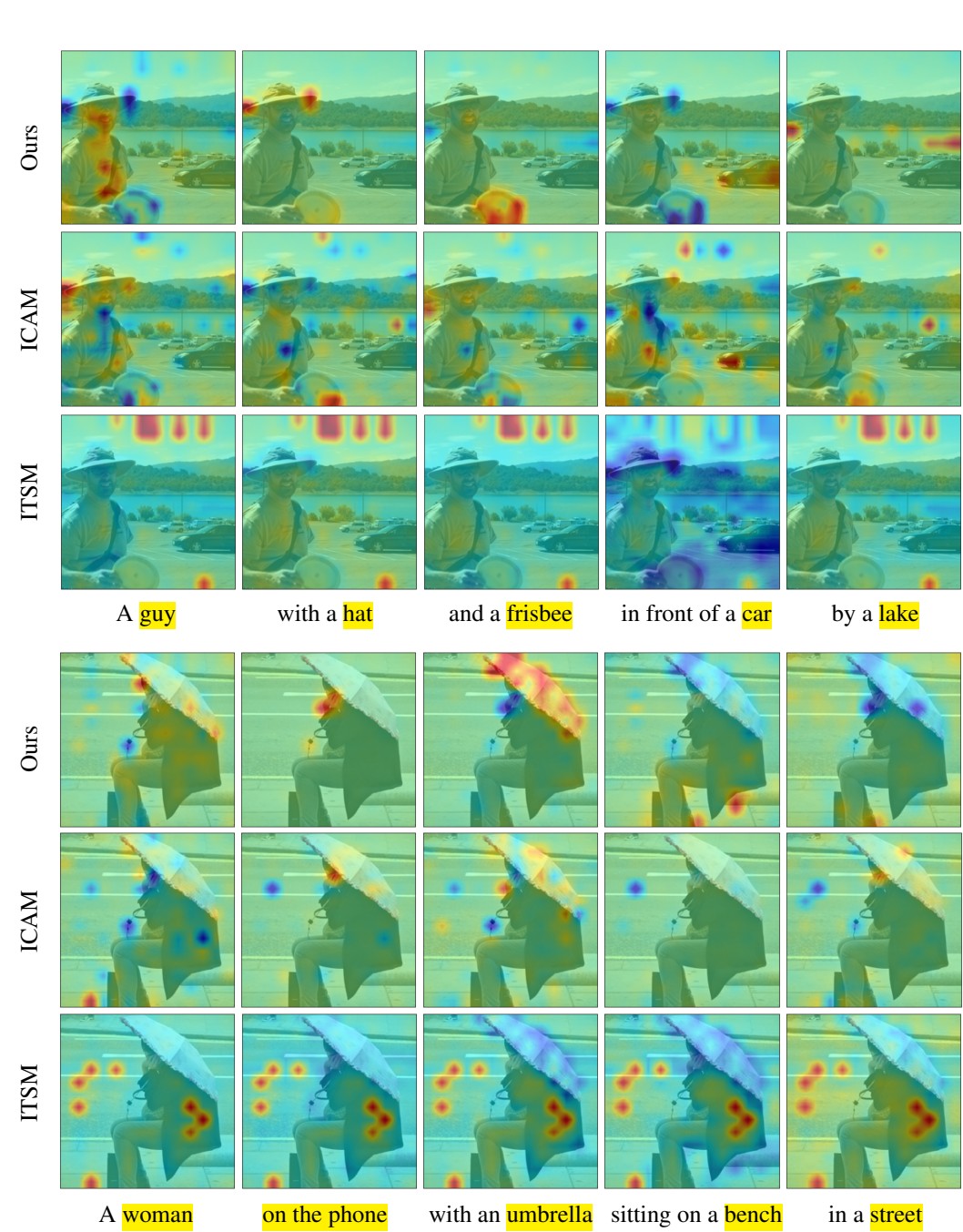

Figure 15: Qualitative comparison between our attributions and the InteractionCAM (ICAM) and ITSM baseline. Heatmaps over images in a given column are for the marked marked parts of the captions in yellow below.

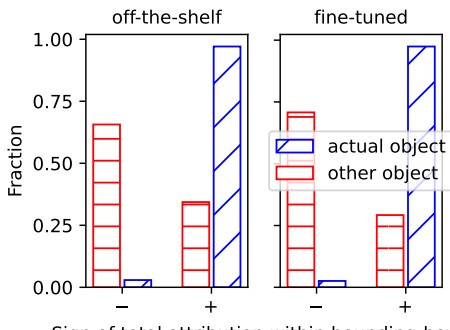

Figure 16: Distribution of signs of attributions to actual and other objects in the image as described in Section 4.2.

| | | COCO | | Flickr30k | |
|---|---|---|---|---|---|
| **Training** | **Method** | mPGE | PGA | mPGE | PGA |
| | ITSM | 18.1 | 21.4 | 19.5 | 23.3 |
| OpenAI | ICAM | 38.6 | 54.6 | 33.5 | 51.4 |
| | Ours | **72.3** | **79.0** | **64.4** | **72.1** |
| | ITSM | 22.8 | 30.3 | 24.5 | 28.7 |
| Laion (tuned) | ICAM | 32.5 | 58.4 | 33.5 | 51.4 |
| | Ours | **71.2** | **83.2** | **56.3** | **63.6** |
| | ITSM | 24.2 | 34.5 | 25.1 | 31.4 |
| DFN (tuned) | ICAM | 33.3 | 46.5 | 24.2 | 42.2 |
| | Ours | **71.4** | **74.6** | **53.1** | **58.3** |
| | ITSM | 25.5 | 38.7 | 26.5 | 33.9 |
| DataComp (tuned) | ICAM | 36.9 | 49.5 | 23.2 | 37.3 |
| | Ours | **72.4** | **75.1** | **50.7** | **60.0** |

Table 6: Localization results of our method compared against the ITSM and InteractionCAM (ICAM) baselines for different models.

Table 7: Class-pairs with the highest average positive attributions between the first class in the caption and the second class in the image (cross-attribution), together with how often the first class appears together with the second one in the COCO labels (co-occurrence).

| Class-pair | Co-occurrence [%] | Cross-attribution [$\times 10^{-3}$] |
|---|---|---|
| Kite – Person | 90.1 | $2.5 \pm 6.9$ |
| Frisbee – Person | 86.9 | $1.9 \pm 3.5$ |
| Surfboard – Person | 95.3 | $1.8 \pm 2.5$ |
| Dining Table – Person | 49.9 | $1.5 \pm 1.7$ |
| Toilet – Sink | 45.6 | $1.0 \pm 2.5$ |

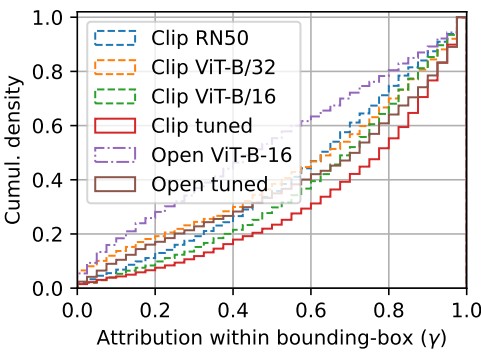 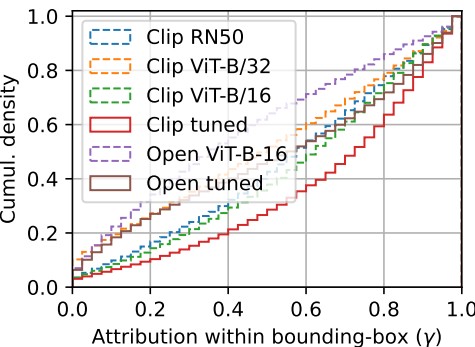

Figure 17: Cumulative $\gamma$-distribution plots of the OpenAI models for the Coco (left) and Flickr30k (right) dataset as described in Section 4.1.

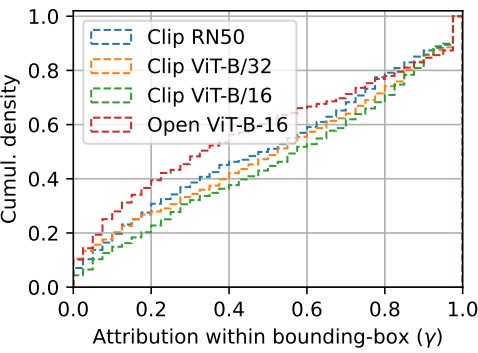 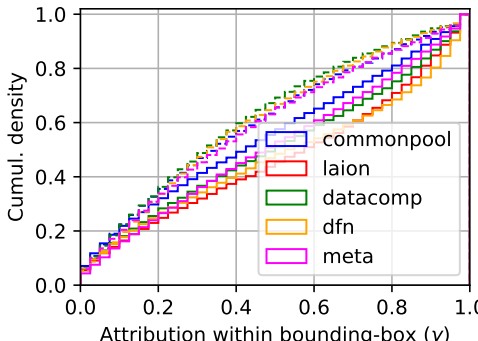

Figure 18: Cumulative $\gamma$-distribution plots for the OpenAI models on HNC (left) and the OpenCLIP models on Flickr30k (right) as described in Section 4.1.

## F  STOCHASTIC DOMINANCE

Stochastic dominance defines an order relation between probability distributions based on their cumulatives. del Barrio et al. (2018) have proposed a significance test building on the principle and Dror et al. (2019) have identified it as being particularly suitable to compare deep neural models. The test's $\epsilon$-parameter is the maximal percentile range where the inferior distribution is allowed to dominate the superior one and Dror *et al.* suggest to set it to $\epsilon < 0.4$. The smaller $\epsilon$, the stricter the criterion. $\alpha$ is the significance level.

## G  RELATION TO GRADCAM

Here, we discuss the relation of integrated gradients Sundararajan et al. (2017) and GradCam. We start by deriving IG for a model $f(\mathbf{a}) = s$ with a vector-valued input $\mathbf{a}$ and a scalar prediction $s$, which might e.g. be a classification score for a particular class. We define the reference input $\mathbf{r}$, begin from the difference between the two predictions and reformulate it as an integral:

$$f(\mathbf{a}) - f(\mathbf{r}) = \int_{\mathbf{r}}^{\mathbf{a}} \frac{\partial f(\mathbf{x})}{\partial \mathbf{x}_i} d\mathbf{x}_i \tag{12}$$

To solve the resulting line integral, we substitute with the straight line $\mathbf{x}(\alpha) = \mathbf{r} + \alpha(\mathbf{a} - \mathbf{r})$ and pull its derivative $d\mathbf{x}(\alpha)/d\alpha = (\mathbf{a} - \mathbf{r})$ out of the integral:

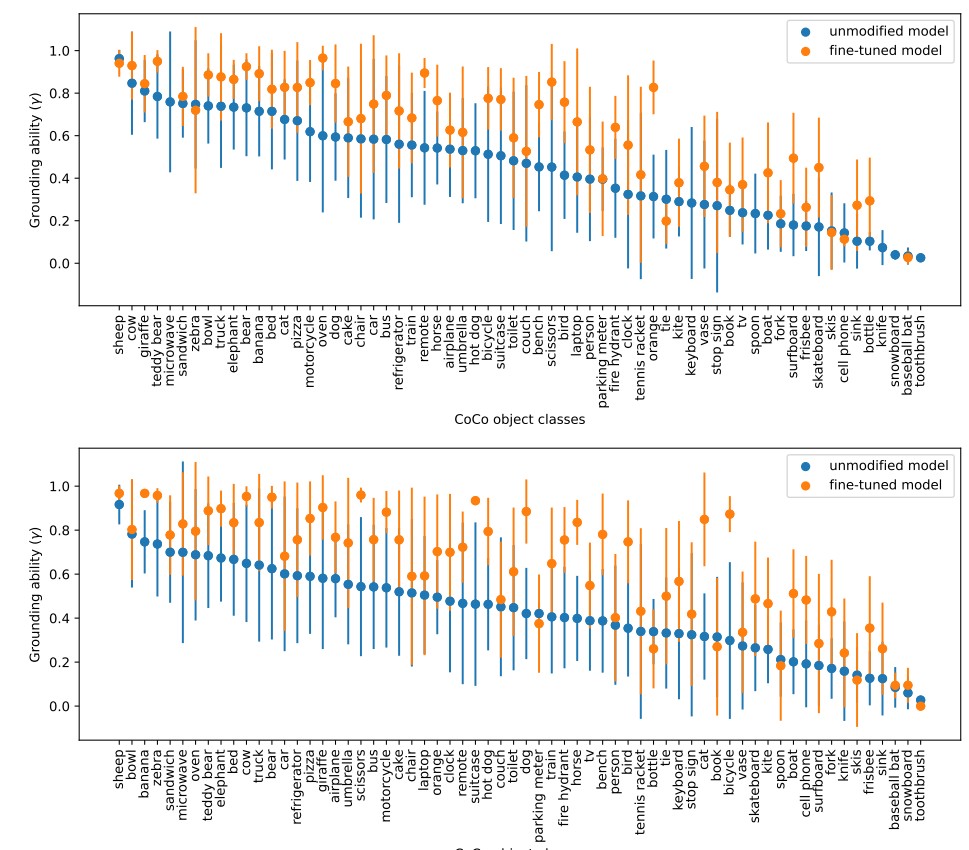

Figure 19: Class-wise evaluation of intrinsic grounding ($\gamma$) in the OpenCLIP Laion (top) and DataComp (bottom) models before and after in-domain fine-tuning as discussed in Section 4.1, cf. Figure 5.

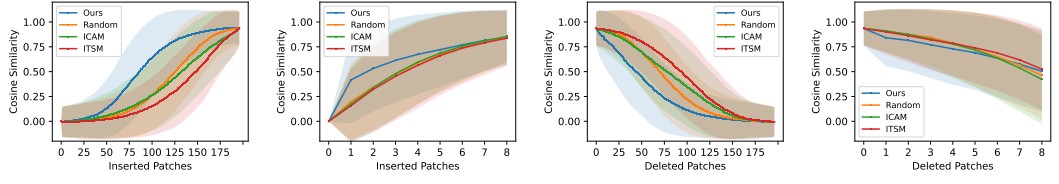

(a) Conditional Image Insertion.

(b) Conditional Text Insertion.

(c) Conditional Image Deletion.

(d) Conditional Text Deletion.

Figure 20: Conditional insertion and deletion performed on either the caption or the image.

$$\int_{\alpha=0}^{1} \frac{\partial f(\mathbf{x}(\alpha))}{\partial \mathbf{x}_i(\alpha)} \frac{\partial \mathbf{x}_i(\alpha)}{\partial \alpha} d\alpha = (\mathbf{a} - \mathbf{r})_i \int_{\alpha=1}^{1} \nabla_i f(\mathbf{x}(\alpha)) \, d\alpha \tag{13}$$

to arrive at the final IG we approximate the integral by a sum over $N$ steps:

$$(\mathbf{a} - \mathbf{r})_i \frac{1}{N} \sum_{n=1}^{N} \nabla_i f(\mathbf{x}(\alpha_n)) \tag{14}$$

If $f(\mathbf{r}) \approx 0$ this is the contribution of feature $i$ in $\mathbf{a}$ to the model prediction $f(\mathbf{a}) = s$. We can now reduce these feature attributions further by setting $N = 1$ and $\mathbf{r} = \mathbf{0}$, to obtain

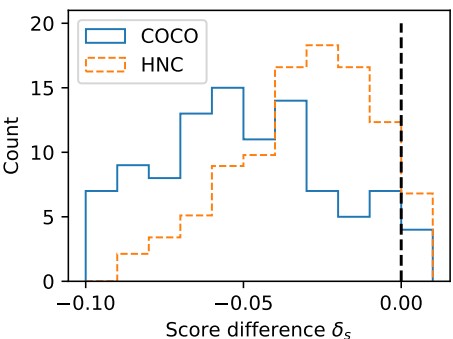 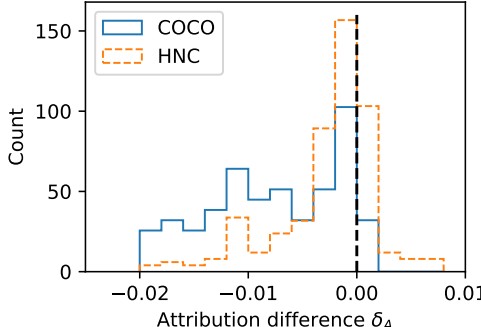

Figure 21: Histograms of the difference between hard negative and original positive captions for the predicted similarity score (left) and the attribution between objects in the caption and corresponding bounding-boxes in the image (right).

$$\mathbf{a}_i \nabla_i f(\mathbf{a}), \tag{15}$$

which is often referred to as *gradient times input* and the basic form of GradCam. The method typically attributes to deep image representations in CNNs, so that $\mathbf{a}$ has the dimensions $C \times H \times W$, the number of channels, height and width of the representation. To reduce attributions to a two-dimensional map, it sums over the channel dimension and applies a relu-activation to the outcome. The original version also average pools the gradients over the spacial dimensions, however, this is technically not necessary.

As discussed earlier, neither integrated gradients nor GradCam can explain dual encoder predictions. Following the logic from above we can, however, derive a "GradCam for similarity" by setting $N = 1$ in the computation of the integrated Jacobians in Equation 9 and using $\mathbf{r}_a = \mathbf{0}$ and $\mathbf{r}_b = \mathbf{0}$. For our attribution matrix from Equation 10 we then receive the simplified version

$$\mathbf{a}_i \frac{\partial \mathbf{g}_k}{\partial \mathbf{a}_i} \frac{\partial \mathbf{h}_k}{\partial \mathbf{b}_j} \mathbf{b}_j. \tag{16}$$

In our experiments we use these attributions as a baseline and call them *Jacobians times Embeddings*. However, setting $N = 1$ is the worst possible approximation to the integrated Jacobians. Therefore, it is not surprising that empirically this version performs worse than our full attributions.

## H  BROADER RELATED WORK

**Metric learning**   refers to the task of producing embeddings reflecting the similarity between inputs (Kaya & Bilge, 2019). Applications include face identification (Guillaumin et al., 2009; Wojke & Bewley, 2018) and image retrieval (Zhai & Wu, 2018; Gao et al., 2014). Siamese networks with cosine similarity of embeddings were early candidates (Chen & He, 2021). The triplet-loss (Hoffer & Ailon, 2015) involving negative examples has been proposed as an improvement but requires sampling strategies for the large number of possible triplets (Roth et al., 2020). Qian et al. (2019) have shown that the triplet-loss can be relaxed to a softmax variant. Sohn (2016) and van den Oord et al. (2019) have proposed the batch contrastive objective which has been applied in both unsupervised (Caron et al., 2020) and supervised representation learning (Khosla et al., 2020) and has lead to highly generalizable image (He et al., 2020) and semantic text embeddings (Reimers & Gurevych, 2019).

**Vision-language models**   process both visual and linguistic inputs. Zhang et al. (2022b) were the first to train a dual-encoder architecture with a contrastive objective on image-text data in the medical domain. With CLIP Radford et al. (2021) have applied this principle to web-scale image captions and the ALIGN model has achieved similar results with alt-text (Jia et al., 2021). In the following, the basic inter-modal contrastive loss has been extended by, intra-modal loss terms (Goel et al., 2022; Lee

et al., 2022; Yang et al., 2022a), self-supervision (Mu et al., 2022), non-contrastive objectives (Zhou et al., 2023), incorporating classification labels (Yang et al., 2022b), textual augmentation (Fan et al., 2023), a unified multi-modal encoder architecture (Mustafa et al., 2022) and retrieval augmentation (Xie et al., 2023). Next to more advanced training objectives, other works have identified the training data distribution to be crucial for performance: Gadre et al. (2023) have proposed the DataComp benchmark focusing on dataset curration while fixing model architecture and training procedure, Xu et al. (2024) have balanced metadata distributions and Fang et al. (2024) propose data filtering networks for the purpose. The strictly separated dual-encoder architecture has been extended to include cross-encoder dependencies (Li et al., 2022a; Pramanick et al., 2023), and multi-modal encoders have been combined with generative decoders (Chen et al., 2023a; Lu et al., 2023; Li et al., 2021; Koh et al., 2023; Alayrac et al., 2022). The CoCa model combines contrastive learning on uni-modal vision- and text-representations with a text generative cross-modal decoder (Yu et al., 2022).

**Visual-linguistic grounding** is the identification of fine-grained relations between text phrases and corresponding image parts (Chen et al., 2023b). Specialized models predict regions over images for a corresponding input phrase (Sadhu et al., 2019; Ye et al., 2019). This objective has been combined with contrastive caption matching (Li et al., 2022b; Datta et al., 2019), and caption generation (Yang et al., 2022c). The VoLTA model internally matches latent image-region and text-span representations (Pramanick et al., 2023). In multi-modal text generative models, grounding has been included as an additional pretraining task (Li et al., 2020; Su et al., 2019; Chen et al., 2020); alternatively grounding abilities can be unlocked with visual prompt learning (Dorkenwald et al., 2024). At the intersect of grounding and explainability, Hendricks et al. (2016) have generated textual explanations for vision models and have grounded them to input images (Hendricks et al., 2018; Park et al., 2018). In this paper, we do not optimize models to explicitly ground predictions, but aim at analyzing to which extent purely contrastively trained dual encoders have this ability.

