# OpenReview forum: "Explaining Vision-Language Similarities in Dual Encoders with Feature-Pair Attributions"
_ICLR.cc/2025/Conference — Submitted to ICLR 2025_

### Official Review · Reviewer_Dy42 · 2024-11-01

**Soundness:** 3
**Presentation:** 3
**Contribution:** 3
**Rating:** 6
**Confidence:** 3

**Summary:**

This paper extends the Integrated Gradients algorithm to explain similarity in CLIP models which consist of two vision-language encoders, via attributing feature pairs (language-vision attribution). This allows capturing of feature interactions and correspondances, rather than having 2 separate attribution entities that are disconnected. Experiments are conducted on a variety of CLIP models from OpenAI and OpenCLIP.

**Strengths:**

- The authors tackle an important topic of interpretation of feature interactions, which is not that well-explored in the literature
- Wide range of CLIP models considered in experiments
- Nice extension of the work "An Attribution Method for Siamese Encoders to CLIP" to the vision-language setting.

**Weaknesses:**

- [W1] There is an abundance of related work that the authors miss. [R1, R2, R3] are papers that are doing exactly the same thing. R1 proposes several baseline methods and evaluation metrics. R3 is also applied to CLIP and shows object-word interactions. The authors did not mention these works, their drawbacks and how their method is better, and did not compare with them. The authors are proposing a method without comparing it to existing methods in the literature. Comparison with baselines (and mentioning how the proposed method is different and what advantages it offers) is an integral part of any research work, which is missing.

- [W2] Related to W1, the authors do not mention neither prove how methods which provide individual feature attributions (e.g., Grad-CAM, Integrated Gradients, LIME...etc) are bad and do not work, and that a new formulation of feature interactions is needed. Simply stating this statement without proving it, is not valid.

- [W3] Poor evaluation: all I see is two graphs in Figure 5 (table 1 is an ablation experiments, not main results). There is no wide enough evaluation metrics and tasks. The only evaluation in Figure 5 is based on bounding box overlaps. However, it is well-agreed upon in XAI that bounding boxes are a very bad way of evaluating explanations, because they assume the model reasoning process aligns with the annotations.  If a model is biased and detects a "hand" rather than a "dumbell", it will be penalized (because the annotated box is for the dumbell), while it should not. Similarly, assume the model can identify a dog solely by its "tail". In this scenario, the model focuses only on physical features of the tail, and disregards other features like the dog's body, fur, or face. A good explanation in this case would highlight the tail. But, the overlap between the bounding box of that explanation (the dog's tail) and the overall bounding box of the dog would be small and penalised (while it should not). These attribution methods act as a way to perform applications such as zero-shot object detection or segmentation (e.g., [R4]), but not as ways to evaluate interpretations.

- [W4] How did the authors arrive to Eq.2? How is is formulated? Based on what? It suddenly appears in this form. Did the authors create this starting point? If so, based on what?

Minor (do not affect my decision):
- The authors mention: "One may expect CLIP models must learn object correspondence between vision and language modes. But to our knowledge, our evaluation is the first piece of evidence indicating that this is actually the case". This is not surprising at all. None of the various applications of CLIP would actually work if this was not the case.
- Not sure what is (top: selections in yellow, bottom: saliencies as above) in Figure 2. Are yellow and red supposed to be corresponding? Or is red the negative?

References\
[R1] Visualizing and Understanding Contrastive Learning, TIP 2023\
[R2] Visualizing deep similarity networks, WACV 2019\
[R3] Model-Agnostic Visual Explanations via Approximate Bilinear Models, ICIP 2023\
[R4] Interpreting CLIP's Image Representation via Text-Based Decomposition, ICLR 2024\

**Questions:**

In general, I feel this paper is not ready for ICLR. The weaknesses greatly outweighs the strength, Therefore, my decision will be, sadly enough, to reject this paper. W1, W2, W3 and W4 are major issues which should be addressed.

---

> ### Author Response · Authors · 2024-11-19
> **Rebuttal**
>
> ### Necessity for feature-interaction attributions [W2]
> *"the authors do not mention neither prove how methods which provide individual feature attributions (e.g., Grad-CAM, Integrated Gradients, LIME...etc) are bad and do not work, and that a new formulation of feature interactions is needed. Simply stating this statement without proving it, is not valid."*
>
> It is a well understood fact that first-order feature attribution methods like GradCAM, IG, LIME, Shap, etc. cannot assess feature interactions. We discuss the related literature in lines 135-144, e.g. [1], [2]. For the case of dual encoders like CLIP, we can see that their predictions only depend on feature interactions by writing out the model definition. \
> $ s = f(\mathbf{a}, \mathbf{b}) = \mathbf{g}(\mathbf{a})^T \mathbf{h}(\mathbf{b}) = \sum_k \mathbf{g}_k (\mathbf{a}) \mathbf{h}_k (\mathbf{b}) $ (cf. Eq. 1, Section 3) \
> All terms contributing to the similarity scores involve multiplicative dependencies between features of the two inputs. There is no contribution of terms involving only features of one input.
> First-order methods can attribute the final similarity scores to either of the inputs, resulting in a single overall attribution map, but they cannot produce attributions for interactions between features.
> ### Related work
> Thank you for pointing out these works. We will make sure to cite them. Please refer to our general response above for a differentiation between these works and our method.
> Generally, with two exceptions, the applied methods fall under first-order attributions and, therefore, do not provide the same level of fine-grained explanations for interactions between captions and images.
> The first exception is the InteractionCAM method in R1. We recognize that this method is equivalent to our Jacobians times embedding (JxE) baseline, which we derive in Appendix F and also refer to as “GradCAM for similarity” in line1204. We will update the name and cite it accordingly to credit that it has been derived prior to our work.
> The second exception is InteractionLIME, proposed in R3, which generalizes LIME to dual-encoder similarity models, by locally approximating the actual CLIP model with a bilinear form. Thus, it explains an approximate version of the real model, which is the central limitation of LIME. Nevertheless, it does provide explainability for feature interactions, and we will credit it accordingly.
>
> ### Theory
> *“How did the authors arrive to Eq. 2”* \
> Our derivation begins with  Eq. 2 and first shows its equality to Eq. 10.
> It then applies the approximations in line 210 to reduce this proven equality to the final formulation of our attributions in Eq. 11.
> We define all variables and terms that appear in Eq. 2 in the previous paragraph (lines 153-160), and explicitly state that Eq. 2 is a rigorous starting point. Additionally, line 162 sketches the subsequent steps of our derivation before going into the details.
> However, we understand that this point in the derivation may cause confusion, and will try to reformulate it to ensure the theory is as accessible as possible. \
> ### References
> [1] Sundararajan et al. The shapley taylor interaction index. PMLR 2020 \
> [2] Janizek et al. Explainaing explanations: Axiomatic feature interactions for deep neural networks. JMLR 2021
> [3] Li et al. Exploring Visual Explanations for Contrastive Language-Image Pre-training. arXiv 2022
> [4] Zhao et al. Gradient-based visual explanations for transformer-based CLIP. PMLR 2024

---

> ### Author Response · Authors · 2024-11-19
> **Clarifications regarding the evaluations**
>
> ### Evaluations
> There are several misunderstandings about our evaluation that we would like to clarify.
>
> *“Table 1 is an ablation experiment, not main results.”* \
> This is not accurate. Table 1 summarizes our bounding-box experiment and is further extended by Tables 2 and 3 in the Appendix, as we could not include all tested models in the main part. Figure 5 visualizes the full distributions of our Gamma metric. Our ablation corresponds to Figure 5 (right). As mentioned above, the JxE baseline corresponds to InteractionCAM from R1, and as discussed with reviewer onpN, our “multiply” baseline is equivalent to the ITSM method from [3]. We will rename this section into “baselines” and cite both methods.
>
> *“The only evaluation in Figure 5 is based on bounding box overlaps”*
> This is also not correct. While Section 4.1 focuses on evaluating the agreement of our attributions with human object annotations, **Section 4.3 performs a second line of experiments based on perturbations**. Here, we first run deletion experiments to evaluate the importance of attributed image patches for the models similarity prediction. This type of experiment is also proposed by [R1], which we will cite. Second, we evaluate the impact of hard negative captions on similarity scores.
> Overall, by evaluating the agreement with human object annotations and conducting perturbation experiments, we follow a common evaluation protocol for visual attributions, as used by, e.g., [4]
> In fact, our Rho metric corresponds to Point-Game accuracy, and our Gamma metric is equivalent to Point Game energy (cf. lines 287-289, 295,-296) in [4]. We will rename these metrics in our evaluation to be consistent with prior work. The reason we do not report additional metrics based on segmentation masks is that our interaction evaluation requires image-caption datasets, which typically only include  bounding-boxes (except for COCO). Unlike Zhao et al. in [4], who evaluate their first-order attributions using image classification datasets, we cannot use such datasets for our approach because we evaluate interactions between full captions and images.
>
> We are aware of the issue you pointed out with the dog example and observe this phenomenon frequently, e.g. the giraffe in Fig. 9, which is identified by its neck. This is precisely the reason why we do not evaluate bounding box overlap. Instead PointGame energy (our gamma) measures the fraction of positive attributions falling inside the bounding box (or mask) as defined in lines 287-288. For your dog example, this metric would correctly be equal to one, as long as the bounding box includes the dog’s tail.

---

> > ### Comment · Reviewer_Dy42 · 2024-11-20
> > **Response to Authors**
> >
> > I thank the authors for their response.
> >
> > > First-order methods can attribute the final similarity scores to either of the inputs, resulting in a single overall attribution map, but they cannot produce attributions for interactions between features.
> >
> > Indeed, after giving it another thought, it is not possible to get an interaction map with first-order methods. I apologize for raising this (wrong) concern.
> >
> > >  Jacobians times embedding (JxE) baseline
> >
> > Thanks for clarifying that this baseline is equivalent to InteractionCAM. In that case, there is a comparison with at least two baselines in the current literature (including ITSM - or the multiply baseline). Regarding InteractionLIME, as the code is not publicly available, I will not ask the authors to compare with it. However, the authors should clearly state the limitations of InteractionLIME in relation to the proposed method (not just mention and cite it - you need to show the limitation of this work, and how the proposed method tackles that limitation)
> >
> > > Evaluation
> >
> > Thanks for the clarification. Please rename all metrics to be consistent with prior work. Otherwise, its hard to realize that the metrics are the same as the ones used in the literature.
> >
> > In general, the authors response addressed my concerns. I have raised my score.

---

> > > ### Author Response · Authors · 2024-11-27
> > > **Note on revision**
> > >
> > > Thank you for acknowledging the second-order characteristic of our method.
> > > We have now uploaded a revised version of the paper, including Figure 1, which directly compares first- and our second-order attributions.
> > > We have also adopted the conditional insertion / deletion framework from [1] for both the image and text input, extending our perturbation evaluation in Section 4.3 to four metrics (cf. Table 3, Figure 6 and 20).
> > > Our baselines in both the perturbation and localization experiment are now InteractionCAM and ITSM. Further, we have changed our evaluation metrics for the localization experiment to PointGame Energy and Accuracy and Table 2 now also includes a quantitative comparison of our method against the baselines, based on these metrics.
> > > With these changes, we hope to address your concerns regarding the extend of our evaluation. Please also refer to our general note on the revision above.
> > >
> > > [1] Sammani et al. Visualizing and understanding contrastive learning. TIP 2023

---

> > > > ### Comment · Reviewer_Dy42 · 2024-11-27
> > > > **response to author**
> > > >
> > > > I thank the authors for the revised manuscript. After going through the revisions in the manuscript, the paper is now in a much better form. Therefore, I choose to increase my score to 6.

---

> > > > > ### Author Response · Authors · 2024-12-02
> > > > > **Acknowledgement**
> > > > >
> > > > > Thank you for acknowledging the revision. We appreciate the active discussion and constructive feedback!

---

### Official Review · Reviewer_vYYt · 2024-11-04

**Soundness:** 2
**Presentation:** 2
**Contribution:** 2
**Rating:** 5
**Confidence:** 4

**Summary:**

This paper is mainly focused on how dual encoder architectures map two types of inputs into one shared embedding space. To this end, authors propose to attribute predictions of any differentiable dual encoder onto feature-pair interactions between inputs. The proposed method shows that CLIP models learn fine-grained correspondences between parts of captions and regions in images. In-domain finetuning can largely improve the visual-linguistic grounding ability.

**Strengths:**

1. The paper is well written and easy to follow.
2. The authors provide extensive illustrations to demonstrate the visual-linguistic alignment results.

**Weaknesses:**

1. While authors demonstrate the improvements by applying the proposed method to CLIP modes, the results seem to be straightforward: (1) visual-linguistic grounding ability heavily varies between object classes and the training data distribution; (2) in-domain finetuning can largely improve such ability.

2. Regarding the results in Table 1, there is one missing baseline which is directly finetuning CLIP models without the proposed method. Such ablated experiment can further show the effectiveness of the proposed method.

3. Why are there some missing results for the HNC dataset in Table 1?

**Questions:**

See the above weakness.

1. The main concern is that there is one missing baseline in Table 1. This baseline should finetune CLIP models without the proposed method to ablate the proposed method.

2. It seems straightforward to get the conclusions that in-domain finetuning can improve the visual-linguistic ability. It is a little bit unclear about the main contributions of this paper.

3. Have you ever compared the T2I and I2T retrieval results w/ and w/o finetuning? Does better visual grounding lead to better retrieval ability?

---

> ### Author Response · Authors · 2024-11-19
> **Rebuttal**
>
> ### Surprisal of Results
> *“The results seem to be straightforward”*
> While this might seem intuitive in hindsight, we do not fully agree, since it over simplifies the results. Our method focuses on fine-grained feature interactions between modalities, allowing us to reveal that, despite extensive pre-training, CLIP models still lack robust grounding abilities for common objects, such as some of the classes in COCO (cf. Section 4.1 § Class-wise evaluation and Figure 6). Our analysis provides clear evidence and detailed insights into these limitations, which had not been demonstrated before.
> ### Baseline
> *“The main concern is that there is one missing baseline in Table 1. This baseline should finetune CLIP models without the proposed method to ablate the proposed method”*
>
> This is a misunderstanding. All models in Table 1, 2 and 3 are unmodified CLIP models and finetuning is only done with the standard contrastive objective (cf. lines 301-303). In fact, our method is a post-hoc explainability method and does not change the model in any way during training or finetuning. Neither do any of the baselines in Section 4.1 § Ablation. The only differences between the models in Table 1, 2, and 3 are their vision architecture (RN50, ViT-B/32, etc.), as well as the pre-training (OpenAI, Laion, DFN, etc.) and finetuning datasets (COCO and Flickr) they are trained on.
> ### Missing results
> The HNC dataset is rather small (500 pairs). Therefore, we only use it as an evaluation dataset and do not train fine-tuned models on it (cf. lines 259-260, 270-271, 299-300).
> This is why we don’t have results for it in the lines where Tuning = Yes in Table 1.
>
> We did not fine-tune on HNC because translating the Image-Text-Matching based data samples into a contrastive learning setting is non-trivial. HNC consists of positive and negative captions constructed from scene graphs. Each positive caption is paired with a negative one, where a small part of the caption (e.g. attributes, relations between objects, or object identities) is altered to create a hard negative.
> In CLIP’s contrastive loss, target labels are constructed from a set matching caption-image pairs within a batch. One solution would be to only use the positive captions. However, this introduces a new issue: we cannot ensure that all other captions in the batch are truly non-matching with the image, as it is possible for a positive caption to also align with other images in the batch.
>
> We did not fine-tune on HNC because translating the Image-Text-Matching based data samples into a contrastive learning setting is non-trivial. HNC consists of positive and negative captions constructed from scene graphs. Each positive caption is paired with a negative one, where a small part of the caption (e.g. attributes, relations between objects, or object identities) is altered to create a hard negative.
> In CLIP’s contrastive loss, target labels are constructed from a set matching caption-image pairs within a batch. One solution would be to only use the positive captions. However, this introduces a new issue: we cannot ensure that all other captions in the batch are truly non-matching with the image, as it is possible for a positive caption to also align with other images in the batch.
> ### Question
> *“Does better grounding ability lead to better T2I/I2T retrieval?”*
>
> This would indeed be an interesting down stream analysis. We are currently in the process of running downstream experiments like this and will consider including them in the revised version.

---

> > ### Author Response · Authors · 2024-11-27
> > **Note on revision**
> >
> > We have now uploaded a revised version of our paper addressing the concerns regarding baselines and the extend of our evaluation. We also added Figure 1 to directly compare our second-order attributions against first-order counter parts.
> > We hope this clarifies the contribution of our work with respect to prior publications. Please also refer to our general note on the revision above.

---

> > > ### Author Response · Authors · 2024-12-02
> > > **Reminder of revision**
> > >
> > > As intended by the ICLR reviewing process, we have uploaded a revised version of our paper. In this revision and our response above, we have addressed the issues that were raised by all reviewers. We would appreciate your feedback on these updates.

---

### Official Review · Reviewer_onpN · 2024-11-04

**Soundness:** 3
**Presentation:** 3
**Contribution:** 3
**Rating:** 6
**Confidence:** 4

**Summary:**

This paper presents a method to interpret CLIP models' cross-modal alignment mechanisms by attributing similarity predictions to specific feature-pair interactions across input modalities (image-text, text-text, image-image). The proposed method leverages integrated gradients to match parts of captions with corresponding regions in images. The authors also show that the model’s grounding abilities vary by object class and are sensitive to the training data distribution, with in-domain fine-tuning improving performance. The paper includes evaluations on datasets with object bounding-box annotations (3.5k image-caption pairs from COCO, 8k pairs from Flickr30k, and 500 pairs from HNC).

**Strengths:**

- This paper is easy to read, the proposed algorithm/methodology is sound, and the technical details are well explained (including a general description for code implementation in Section 7).

- This paper is an extension of previous work for language-only Siamese models in NLP (Möller et al., 2023; 2024) -- it provides a principled implementation for assessing the visual-linguistic grounding abilities of CLIP models, showing that the proposed approach in [1, 2] generalizes to vision and language models (particularly, image-text and image-image matching).

- The proposed generalization of integrated gradients for object attribution is simple yet effective, and seems to work well for CLIP models.


[1] Lucas Möller, Dmitry Nikolaev, and Sebastian Padó. An attribution method for siamese encoders. In Proceedings of EMNLP, Singapore, 2023.

[2] Lucas Möller, Dmitry Nikolaev, and Sebastian Padó. Approximate attributions for off-the-shelf
Siamese transformers. In Yvette Graham and Matthew Purver (eds.), Proceedings of the 18th
Conference of the European Chapter of the Association for Computational Linguistics (Volume 1:
Long Papers), pp. 2059–2071, St. Julian’s, Malta, March 2024. Association for Computational
Linguistics.

**Weaknesses:**

*"One may expect CLIP models must learn object correspondence between
vision and language modes. But to our knowledge, our evaluation is the first piece of evidence
indicating that this is actually the case."* -- this is a strong claim that does not hold. A large number of prior works has systematically explored gradient-based methods that uncover the grounding capabilities of CLIP (and other visual-language models) [3, 4, 5, 6].

- Limited evaluation: gradient-based explanations are typically evaluated in pointing game accuracy, segmentation masks, object detection and qualitative comparison against prior work. This work only compares the grounding ability in bounding-box evaluation. It is unclear how the proposed approach fares against prior relevant work.

- Due to limited benchmarks and testbeds, it is difficult to determine if the proposed method is competitive against prior work, and to what extent it provides significant new contributions.

[3] Chenyang ZHAO, Kun Wang, Xingyu Zeng, Rui Zhao, & Antoni B. Chan (2024). Gradient-based Visual Explanation for Transformer-based CLIP. In Forty-first International Conference on Machine Learning.

[4] Yossi Gandelsman, Alexei A Efros, & Jacob Steinhardt (2024). Interpreting CLIP's Image Representation via Text-Based Decomposition. In The Twelfth International Conference on Learning Representations.

[5] Weĳie Tu, Weijian Deng, & Tom Gedeon (2023). A Closer Look at the Robustness of Contrastive Language-Image Pre-Training (CLIP). In Thirty-seventh Conference on Neural Information Processing Systems.

[6] Bousselham, Walid, et al. "LeGrad: An Explainability Method for Vision Transformers via Feature Formation Sensitivity." arXiv preprint arXiv:2404.03214 (2024).

**Questions:**

- As an extension of previous work, the contribution seems incremental due to the limited experimental analysis. Is it possible to show experimental results with other VLM architectures?

- Results in Figure 5 are difficult to parse. Is it possible to separate the results into different tables per model (to show in the Appendix)?

_____

Not questions to answer, but suggestions that don't need to be addressed:
- What are the results after finetuning with out-of-domain data? -- this might give a better picture and analysis of the models' performance and explainable approach

- In Related Work (Sec. 2), there is a rather extensive and detailed explanation of vision-language models and their training objectives, however,  this work only focuses on CLIP models -- it might be interesting to correlate how different learning objectives/models correlate with the proposed post-hoc evaluation, and how they impact the grounding/explanation capabilities exposed by prior work and the proposed approach.

---

> ### Author Response · Authors · 2024-11-19
> **Rebuttal**
>
> ### Prior Work
> *“A large number of prior works has systematically explored gradient-based methods that uncover the grounding capabilities of CLIP”
> “it is difficult to determine if the proposed method is competitive against prior work”*
>
> Previous work on gradient-based attributions for CLIP mostly focuses on first-order methods which *cannot* explain interactions between captions and images. This includes the cited papers [3] and [6]. Please also refer to our general response above, addressing this aspect. [4] and [5] analyze CLIP models but do not focus on attributions for similarity predictions. We will nevertheless make sure to incorporate them into our related work.
> In contrast, our method produces second-order attributions that can explain which parts of a caption correspond to which regions in an image (cf. lines 39-45, 47-49, 224-225). Thus, our attributions are more fine-grained than previous work, and are not directly comparable to these first-order methods.
> However, we do compare against two second-order baselines in Section 4.1 § Ablation and Section 4.3 § Image patch removal. In the meantime we have realized that our multiply baseline is equivalent to the ITSM method from [4] and our JxE baseline corresponds to the InteractionCAM from [5] and we will cite them accordingly.
> ### Evaluation
> *“gradient-based explanations are typically evaluated in pointing game accuracy, segmentation masks, object detection and qualitative comparison against prior work”*
>
> Our Rho metric actually corresponds to PointGame accuracy and our Gamma is equivalent to PointGame energy as it is defined by [3] (cf. lines 287-290, 295-296)). We will rename these metrics to be consistent with previous literature. We do not use segmentation masks because they are not available for Flickr30k and HNC but instead base our evaluation on bounding-boxes which are available for all three captioning datasets including COCO.
> As we evaluate interactions between full captions and images, we cannot use image classification datasets like ImageNet as some first-order attribution methods have done, e.g. [3].
> Adding qualitative comparisons to ITSM and InteractionCAM will be straightforward and we will do so in a revised version.
> ### References
> [1] Li et al. Exploring visual explanations for contrastive language-image pre-training. arXiv \
> [3] Zhao et al. Gradient-based visual explanations for transformer-based CLIP. PMLR 2024 \
> [4] Li et al. Exploring Visual Explanations for Contrastive Language-Image Pre-training. arXiv 2022 \
> [5] Sammani et al. Visualizing and understanding contrastive learning. TIP 2023

---

> ### Author Response · Authors · 2024-11-22
> **Questions**
>
> Sorry for the late reply regarding your questions.
>
> *“Is it possible to show experimental results with other VLM architectures?”* \
> Our attribution method is limited to contrastive models. While it does generalize to different dual encoder models, e.g. for different data modes like audio, it does not generalize to arbitrary VLM architectures. This is because in Eq. 5 we make explicit use of the dual encoder architecture. As a result, we can separate derivatives and integrals in Eq. 6 and proceed as shown. This step would not be possible if there were any dependencies between the two encoders as is the case in e.g. cross-attention-encoders. The reason why in our related work section we cite a number of generative (non-contrastive) VLMs is that they often re-use CLIP encoders in their architectures. \
>
> *"Results in Figure 5 are difficult to parse. Is it possible to separate the results into different tables per model (to show in the Appendix)?"* \
> Thank you for pointing this out. Figure 5 shows cumulative distributions of the Point Game Energy metric (previously our gamma), for which we report median values in Tables 1 and 2 (and the appendix). We intended to also show distributions rather than one summary statistics alone to give the full picture. The cumulative attributions also allow us to compare several models at one glance. Intuitively, “the further the distribution comes to the bottom right the better”. Thus, Figure 5 shows that the fine-tuned models largely improve over the shelf models (left) and our attribution method performs much better than the baselines (right). Finally, the significance test that we use is based on these cumulative distributions (cf. lines 303-306 and Appendix F). We will try to describe these aspects better in the revision.

---

> ### Comment · Reviewer_onpN · 2024-11-26
>
> I would like to thank the authors for the clarifications and acknowledging prior work!
>
> I am still a bit confused regarding the proposed baselines and existing work. I strongly suggest the authors update the revised paper; I think reading the revised version would greatly help the reviewers and authors for a fluent and better discussion.

---

> > ### Author Response · Authors · 2024-11-27
> > **Note on revision**
> >
> > We have now uploaded a revised version of the paper. Following your suggestion, we have changed the metrics of our localization evaluation to Point Game Energy and Accuracy.
> > Figure 15 (in the Appendix) now includes qualitative comparisons between our method and the InteractionCAM as well as the ITSM methods. Table 2 quantifies the performance of all three methods within the PG framework.
> > Additionally, we have extended the perturbation experiment to the conditional insertion / deletion framework for contrastive models proposed by [1]. Table 3 reports the resulting four AUC metrics for our method and the baselines.
> > In order to visualize the difference between our second-order attribution and first-order equivalents we have added Figure 1.
> > With these changes, we hope to address your concerns regarding the evaluation and prior work. Please also refer to our general response above.
> >
> > [1] Sammani et al. Visualizing and understanding contrastive learning. TIP 2023

---

> > > ### Comment · Reviewer_onpN · 2024-11-28
> > >
> > > Thank you for uploading the revised version. I think it is now clearer to read and easier to process the strengths of this paper. Both Figures 1 and 2 feel complementary. I also thank the authors for adding relevant experimental results to validate their approach. I'm increasing my rating accordingly.

---

> > > > ### Author Response · Authors · 2024-12-02
> > > > **Acknowledgement**
> > > >
> > > > Thank you for acknowledging our revision. We appreciate the constructive feedback!

---

### Official Review · Reviewer_jo8w · 2024-11-04

**Soundness:** 3
**Presentation:** 3
**Contribution:** 2
**Rating:** 5
**Confidence:** 3

**Summary:**

This paper investigates the mechanisms by which dual encoder architectures, such as CLIP models, compare two types of inputs (e.g., text and images) by mapping them into a shared embedding space and learning their similarities. To address the lack of understanding of how these models compare inputs, the authors present two main contributions:

Derivation of Feature-Pair Attribution Method: The authors derive a method to attribute the predictions of any differentiable dual encoder onto the feature-pair interactions between its inputs. This method can explain interactions between inputs of any differentiable dual encoder model without requiring modifications to the trained model.
Application to CLIP Models: The authors apply their attribution method to CLIP-type models and demonstrate that these models can learn fine-grained correspondences between parts of captions and regions in images. They show that the models can match objects across input modes and account for mismatches. The visual-linguistic grounding ability of these models varies significantly between object classes, depends on the training data distribution, and improves with in-domain training.

The paper includes various experiments to validate the proposed method, such as:
Evaluating object bounding-box attributions to systematically assess the visual-linguistic grounding abilities of dual encoders.
Analyzing attributions to other objects to identify negative attributions for mismatched objects.
Creating and evaluating hard negative captions to test the model's response to errors in captions.

**Strengths:**

1. The paper is well-written and clearly structured.

2. The mathematical derivations and equations are clearly presented (But I did't check carefully)

3. The proposed feature-pair attribution method has the potential to significantly advance the understanding of how dual encoder models, particularly vision-language models, compare inputs and make predictions. This understanding is crucial for improving model interpretability and trustworthiness.

**Weaknesses:**

1. The authors acknowledge that their feature-pair attribution method is an approximation and may not fully capture the exact contributions of feature interactions. This approximation introduces potential inaccuracies in the interpretation of attributions.

2. The experiments and evaluations are primarily focused on CLIP models and specific datasets (e.g., COCO, Flickr30k, HNC). Conducting experiments on a broader range of dual-encoder models and datasets from different domains (e.g., medical imaging, audio-visual data) could demonstrate the generalizability and versatility of the method.

3. While the paper introduces a promising analysis method for attributing predictions of dual encoder models to feature-pair interactions, it lacks a straightforward way to quantify the performance of existing models using this method.
The absence of a standardized metric, similar to FID in image generation or BLEU [1] in machine translation, limits the broader applicability and impact of the method. Having such a metric could significantly enhance the influence and usability of the proposed method.

4. The paper primarily focuses on analyzing and explaining existing phenomena within CLIP models rather than providing solutions or improvements to the training process. While this analysis is valuable, it leaves several important aspects of CLIP training unexplained.
For example, the paper does not address why CLIP models require an extremely large batch size during training, which is a critical aspect of their performance and efficiency.

[2] [Bleu: a Method for Automatic Evaluation of Machine Translation](https://aclanthology.org/P02-1040) (Papineni et al., ACL 2002)

**Questions:**

None

---

> ### Author Response · Authors · 2024-11-19
> **Rebuttal**
>
> ### Scope and limitation of local attributions
> All existing post-hoc explainability methods involve a form of simplification or approximation. Essentially, the only complete explanation for a model are the model parameters themselves, which is however, not tangible.
> Therefore, it is important to clearly define which aspects of a model an explainability model can explain. Local attribution methods like ours can explain which input features are important for a given prediction. They can e.g. not explain why this is the case.
> This is also the reasoning why our method cannot explain hyperparameter settings like the large required batch-size in CLIP models.
> In [1], Murdoch et al. provide an elaborate discussion of these trade-offs in explainability research.
> ### Focus on CLIP
> While our derivation in Section 3 does generalize to any differentiable dual encoder, we focus our experiments on CLIP models in the vision-language domain in this work. This is because they are a well known example of multimodal dual encoders and there is already a lot to evaluate in these models. We absolutely agree that models in different domains like audio would be interesting but leave for future work at this stage.
> ### A standardized metric
> Evaluation is generally a challenge in the XAI domain and it is difficult to establish standardized metrics like BLEU comparing against ground truth, because there is no ground truth for explanations. With our experiments, we stick to a common protocol of evaluating visual attributions: Section 4.1 evaluates whether the model’s attributions align with human annotations, and the perturbation experiments in Section 4.3 test whether attributed regions are relevant for the similarity prediction. This procedure closely aligns with e.g. [2], whose point-game accuracy corresponds to our Rho and their point-game energy is equivalent to our gamma metric. We will consider renaming our metrics to be consistent with prior literature.
> ### References
> [1] Murdoch et al. Definitions, methods, and applications in interpretable machine learning. PNAS 2019 \
> [2] Zhao et al. Gradient-based visual explanations for transformer-based CLIP. PMLR 2024

---

> > ### Author Response · Authors · 2024-11-27
> > **Note on revision**
> >
> > To be consistent with previous work, our localization experiment now reports the point-game energy and and point-game accuracy metrics (cf. Table 1 and 2 in the revised version) that were also used in e.g. [1]. Additionally, we have extended our perturbation experiment to the conditional deletion and insertion framework proposed by [2], which extends this evaluation to four metrics.
> > Please also refer to our general note on the revision to all reviewers above.
> > With these changes, we hope to address your concern about the standardization of our evaluation.
> >
> > [1] Zhao et al. Gradient-based visual explanations for transformer-based CLIP. PMLR 2024
> > [2] Sammani et al. Visualizing and understanding contrastive learning. TIP 2023

---

> > > ### Author Response · Authors · 2024-12-02
> > > **Reminder of revision**
> > >
> > > As intended by the ICLR reviewing process, we have uploaded a revised version of our paper. In this revision and our response above, we have addressed the issues that were raised by all reviewers. We would appreciate your feedback on these updates.

---

### Author Response · Authors · 2024-11-19
**Response to all**

We thank all reviewers for their time and feedback.
To begin with, we would like to make an important clarification regarding the novelty of our study with respect to the related work that two reviewers have pointed out. While all of these works contribute to a better understanding of CLIP (and other contrastive image encoders) through feature attributions, there is an important distinction between their approaches and ours. The cited papers (almost) all apply first-order attribution methods to attribute the predicted similarity score to either the input image or the caption. This results in one single attribution map for a given image-caption pair or a class-label alternatively.
In contrast, our method provides second-order attributions enabling the evaluation of all possible interactions between a caption and an image. We can analyze which parts of a full caption correspond to which regions in a given image (cf. Eq. 10/11 and lines 39-45, 47-49, 207-209, 222-227).
The fact that first-order attribution methods are insufficient to assess feature interactions is well-established in the literature in both the ML and NLP community and we discuss the related work in lines 136-144, including [1, 2, 3].

Throughout our paper, we visualize our feature-pair attributions for interactions by highlighting spans in captions and bounding boxes in images in yellow, and by displaying heat maps over the respective second mode. These heat maps differ noticeably depending on the selected elements, as we show qualitatively in Figure 1 and evaluate quantitatively against human bounding-box annotations (Section 4.1). Additionally, we show that they are largely complementary for different selections (Section 4.2). The ability to assess such fine-grained correspondence between two modalities processed by dual encoders, e.g. captions and images in CLIP models, without modifying the model itself, its gradients, or embeddings is a novel contribution.
This, and only this, is what we are referring to in lines 489-490: “...to our knowledge, our evaluation is the first piece of evidence indicating that this [the fine-grained correspondence between objects in captions and images] is actually the case.” We will ensure to formulate this more precisely in the revised version.

### References
[1] Sundararajan et al. The shapley taylor interaction index. PMLR 2020 \
[2] Janizek et al. Explainaing explanations: Axiomatic feature interactions for deep neural networks. JMLR 2021
[3] Levy et al. Do supervised distributional methods really learn lexical inference relations? NAACL 2015

---

### Author Response · Authors · 2024-11-27
**Note on revision**

We thank all reviewers for their constructive feedback again.
We tried to take all of it into account and created a revised version of the paper.
The most important changes include:
- We changed our metrics in the object localization experiment to Point Game Accuracy (PGA) and Paint Game Energy (PGE)
- We included the InteractionCAM and ITSM methods as baselines, and report localization metrics for both in Table 2
- Additionally, we show qualitative comparisons between our method and the two baselines in Figure 15
- We extended the perturbation experiments to the conditional insertion/deletion framework proposed by Sammani et al. and evaluate both the image and text side. AUC metrics for the four resulting cases are included in Table 3. The corresponding perturbation insertion and deletion figures are displayed in the appendix in Figure 20.
- We updated our related work concerning the papers that were pointed out.
- We created a new plot (Figure 1) demonstrating the difference between our second-order feature-interaction attributions and their first-order counterparts.

---

### Meta-Review · Area_Chair_uVAs · 2024-12-22

**Metareview:**

This paper explores how dual encoder architectures, such as CLIP models, align text and images in a shared embedding space. The authors propose a feature-pair attribution method using gradients to explain predictions by attributing them to interactions between input features (tokens & pixels). Applying this method to CLIP models reveals fine-grained correspondences between image regions and caption parts, with grounding abilities varying by object class and improving with in-domain fine-tuning. The method is validated through experiments on datasets with bounding-box annotations and tests with hard negative captions. The paper is well-written and provides several qualitative results and contains several type of CLIP models that are explored. The paper's weaknesses are the lack of discussion and comparison of several prior vision-language attribution methods and on more diverse datasets, lack of clear evaluation metrics, and no provision of a solution that does improve e.g. the localisation ability.
Some points were addressed during the discussion period, but overall the paper needs further refinement to meet the high bar of acceptance of ICLR and the AC thus recommends rejection.

**Additional Comments On Reviewer Discussion:**

Reviewers Dy42 and onpN engaged in back-and-forth discussion with the authors, addressing points such as further clarity on comparing first-order vs second-order attribution methods, the inclusion of further baselines and updates to the related works.
Overall these points clearly improved the paper, yet to only moderate excitement from the reviewers (5,5,6,6 overall). The remaining weaknesses need further refinement. The AC hopes the authors incorporate the feedback from all reviewers and resubmit to another venue.

---

### Decision · Program_Chairs · 2025-01-22

Reject